# Nutrient sensitive protein *O*-GlcNAcylation modulates the transcriptome through epigenetic mechanisms during embryonic neurogenesis

Shama Parween[1], Thilina T Alawathugoda[1], Ashok D Prabakaran[2], S Thameem Dheen[3], Randall H Morse[4], Bright Starling Emerald[2,5], Suraiya A Ansari[1,5]

**Protein *O*-GlcNAcylation is a dynamic, nutrient-sensitive mono-glycosylation deposited on numerous nucleo-cytoplasmic and mitochondrial proteins, including transcription factors, epigenetic regulators, and histones. However, the role of protein *O*-GlcNAcylation on epigenome regulation in response to nutrient perturbations during development is not well understood. Herein we recapitulated early human embryonic neurogenesis in cell culture and found that pharmacological up-regulation of *O*-GlcNAc levels during human embryonic stem cells' neuronal differentiation leads to up-regulation of key neurogenic transcription factor genes. This transcriptional de-repression is associated with reduced H3K27me3 and increased H3K4me3 levels on the promoters of these genes, perturbing promoter bivalency possibly through increased EZH2-Thr311 phosphorylation. Elevated *O*-GlcNAc levels also lead to increased Pol II-Ser5 phosphorylation and affect H2BS112*O*-GlcNAc and H2BK120Ub1 on promoters. Using an in vivo rat model of maternal hyperglycemia, we show similarly elevated *O*-GlcNAc levels and epigenetic dysregulations in the developing embryo brains because of hyperglycemia, whereas pharmacological inhibition of *O*-GlcNAc transferase (OGT) restored these molecular changes. Together, our results demonstrate *O*-GlcNAc mediated sensitivity of chromatin to nutrient status, and indicate how metabolic perturbations could affect gene expression during neurodevelopment.**

## Introduction

Cellular metabolism could play significant roles in gene expression regulation through epigenetic mechanisms (Lu & Thompson, 2012; Xu et al, 2016; Sharma & Rando, 2017). The enzymes responsible for DNA or histone protein modifications such as histone acetyltransferases, histone methyltransferases and histone deacetylases (HDACs) use cellular metabolites as substrates or cofactors. Thus, nutrient abundance/scarcity and activation of specific metabolic pathways could alter the epigenome depending upon the nutrient status of the cell (Reid et al, 2017).

As epigenetic mechanisms play key roles in stem cell fate determination during development and later differentiation, the functions of cellular metabolites in epigenetic regulation of stem cell fate determination need to be thoroughly investigated. This could also help to uncover the pathological mechanisms associated with nutrient abundances, such as hyperglycemia/hyperlipidemia or metabolic reprogramming during tumorigenesis. Nutrient-sensing molecules on the cell surface and inside sense and signal nutrient abundance or scarcity to the cell so that the cellular metabolic pathways could adjust the production or utilization of metabolites accordingly (Efeyan et al, 2015). As mentioned above, nutrient abundance or scarcity is also perceived by chromatin through different DNA or histone-modifying enzymes. One such nutrient-sensing pathway is the hexosamine biosynthesis pathway which sits at the centre of all major metabolic pathways including glucose, fatty acid, amino acid, and nucleotide metabolisms and uses substrates from these pathways leading to the production of uridine diphosphate N-acetylglucosamine (UDP-GlcNAc) (Love & Hanover, 2005; Hanover et al, 2010). UDP-GlcNAc is used as the substrate for *O*-GlcNAc modification of serine/threonine amino acids on diverse cellular, nuclear and mitochondrial proteins (Love & Hanover, 2005). A pair of enzymes, *O*-GlcNAc transferase (OGT) and *O*-GlcNAcase (OGA), are responsible for adding and removing the *O*-GlcNAc moiety from proteins. Intriguingly, metabolites such as acetyl-coA, S-adenosyl methionine or α-ketoglutarate, which are used by histone acetyltransferases, histone methyltransferases, or HDACs, respectively, are primarily used in either energy production or as cellular building blocks in biosynthetic reactions. On the other hand, UDP-GlcNAc is exclusively used for protein modification thus primarily playing role in

[1]Department of Biochemistry and Molecular Biology, College of Medicine and Health Sciences, United Arab Emirates University, Al Ain, United Arab Emirates [2]Department of Anatomy, College of Medicine and Health Sciences, United Arab Emirates University, Al Ain, United Arab Emirates [3]Department of Anatomy, Yong Loo Lin School of Medicine, National University of Singapore, Singapore, Singapore [4]New York State Department of Health, Wadsworth Center, Albany, NY, USA [5]Zayed Center for Health Sciences, United Arab Emirates University, Al Ain, United Arab Emirates

Correspondence: sansari@uaeu.ac.ae

nutrient signalling. Thus, hexosamine biosynthesis pathway being central to all major metabolic pathways functions as a key nutrient sensor and signal transducer. Among the list of proteins identified as *O*-GlcNAc modified are gene specific transcription factors (TFs), epigenetic regulators and histone proteins (Hart, 2019; Sheikh et al, 2021). Therefore, *O*-GlcNAcylation by serving as a potential nutrient sensor could transfer information of cellular metabolic status to the chromatin and regulate gene expression. Hence, it is not surprising that several pathologies associated with a perturbed cellular metabolism, including diabetes, cancer and neurodegenerative diseases, show perturbed protein *O*-GlcNAcylation (Copeland et al, 2013; Yang & Suh, 2014). In addition, mutations in OGT protein are associated with neurodevelopmental disorders and intellectual disability (Pravata et al, 2019, 2020).

Recent studies have also uncovered the roles of protein *O*-GlcNAcylation in various stem cell functions (Na et al, 2020; Sheikh et al, 2021). *O*-GlcNAc modification of pluripotency related TFs such as octamer-binding transcription factor 4 (OCT4), sex-determining region Y-box 2 (SOX2), and NANOG are shown to be responsible for embryonic stem cells (ESCs') maintenance of pluripotency (Jang et al, 2012). Direct *O*-GlcNAcylation and/or association of OGT with several lineage-specific TFs and chromatin regulators such as enhancer of zeste homolog 2 (EZH2), SIN3A and HDACs are associated with cell fate determination during development and disease (Dehennaut et al, 2014; Leturcq et al, 2017). In addition, OGT forms complexes with ten-eleven translocation (TETs) and other chromatin regulators, including SIN3A and nucleosome remodeling deacetylase complex (Zhang et al, 2014), leading to *O*-GlcNAcylation of chromatin. In ESCs, the association of TET1 with OGT leads to trimethylation of H3K4 (H3K4me3) (Vella et al, 2013). Core histone proteins, H3, H4, H2A, and H2B are also found to be *O*-GlcNAc modified and shown to be involved in cell cycle regulation and cell fate decisions, thus adding to the complexity of *O*-GlcNAc mediated gene regulation (Sakabe et al, 2010). In addition to increasing H3K4me3 levels, it is also reported that *O*-GlcNAcylation down-regulates H3K9me3 levels. Moreover, *O*-GlcNAcylation of H2B on Serine112 (H2BS112*O*-GlcNAc) promotes the transcriptional activation mark H2B-monoubiquitination on K120 (H2BK120Ub1) (Fujiki et al, 2011). These results suggest that *O*-GlcNAcylation could also indirectly regulate other histone modifications.

Therefore, it is relevant to determine whether physiological changes in nutrient abundance contribute to altered cell fate decisions during embryonic development. Maternal chronic hyperglycemia has been implicated in defects of embryonic development and adverse developmental outcomes in the offspring. Epidemiological data and previous experimental studies using animal models have suggested defects in neurodevelopmental outcomes in the offspring resulting from neuroanatomical changes during gestation (Márquez-Valadez et al, 2018; Kong et al, 2020). However, molecular mechanisms behind such defects, including the role of protein *O*-GlcNAcylation, are not well known. Moreover, although the role of protein *O*-GlcNAcylation in epigenome regulation is well studied, epigenetic dysregulation due to sustained *O*-GlcNAcylation during embryonic neurogenesis is not clear. It is also unknown whether the regulation of protein *O*-GlcNAcylation due to nutrient abundance can affect histone modifications to regulate gene expression during neurodevelopment and differentiation. To

answer these questions, we have used human embryonic stem cells (hESCs') differentiation into cortical neurons as a model and shown that sustained *O*-GlcNAc levels during in vitro corticogenesis led to early and increased neuronal differentiation. We also found that the expression of several lineage-specific TFs was up-regulated in the presence of elevated *O*-GlcNAc levels (Parween et al, 2017). Therefore, in this study, we built upon these findings and attempted to determine how increased protein *O*-GlcNAcylation affects the whole transcriptome during neural differentiation of hESCs and to identify the critical epigenetic mechanisms responsible for gene expression changes. Moreover, we have also used a rat model of maternal hyperglycemia to understand the physiological relevance of *O*-GlcNAc-mediated gene regulatory mechanisms on neurodevelopment in vivo.

Using these two model systems, we provide evidence that sustained *O*-GlcNAc levels during hESCs-neuronal differentiation in cell culture affect histone modifications and alter gene expression during neurogenesis. We observe similar effects due to hyperglycemia in animals during pregnancy, which elevated global *O*-GlcNAc levels in developing embryo brains. In particular, increased global *O*-GlcNAc levels in hESCs during neural differentiation resulted in transcriptional up-regulation of several neurogenic TF genes. This transcriptional activation was associated with increased H3K4me3 levels and decreased H3K27me3 levels on the promoters of these genes. The decrease in H3K27me3 may have resulted from increased levels of EZH2-Thr311 phosphorylation observed due to elevated *O*-GlcNAc levels during neural differentiation. EZH2-Thr311p was previously reported to inhibit the catalytic activity of EZH2 of polycomb repressive complex 2, leading to reduced H3K27me3 (Wan et al, 2018). The changes in gene expression due to elevated *O*-GlcNAc levels were also associated with increased Pol II Ser5-p levels suggesting increased Pol II transcription initiation on these genes. Interestingly, similar molecular changes were also observed in the developing brains of hyperglycemic rat embryos, suggesting that hyperglycemia through elevated *O*-GlcNAc levels could alter embryonic neurogenesis.

## Results

### Whole-genome transcriptome analysis identifies genes expressed differentially due to sustained *O*-GlcNAcylation during in vitro embryonic cortical neurogenesis

We have previously shown that increased levels of *O*-GlcNAc affect cortical neurogenesis of hESCs inducing early neuronal differentiation and affecting the expression of lineage-specific TFs (Parween et al, 2017). Here, we investigate further the effects of elevated *O*-GlcNAc levels on gene expression during neural differentiation of hES cells. We have performed high-throughput RNA-seq analysis using hES cells from four stages of in vitro neural differentiation to achieve a temporal gene expression profile of human corticogenesis and to understand the gene expression changes due to sustained *O*-GlcNAcylation in a temporal manner. We have adapted an in vitro cortical embryonic neurogenesis method of hESCs originally published by Chambers et al (Chambers et al, 2009; van de Leemput et al, 2014) and used in our previous

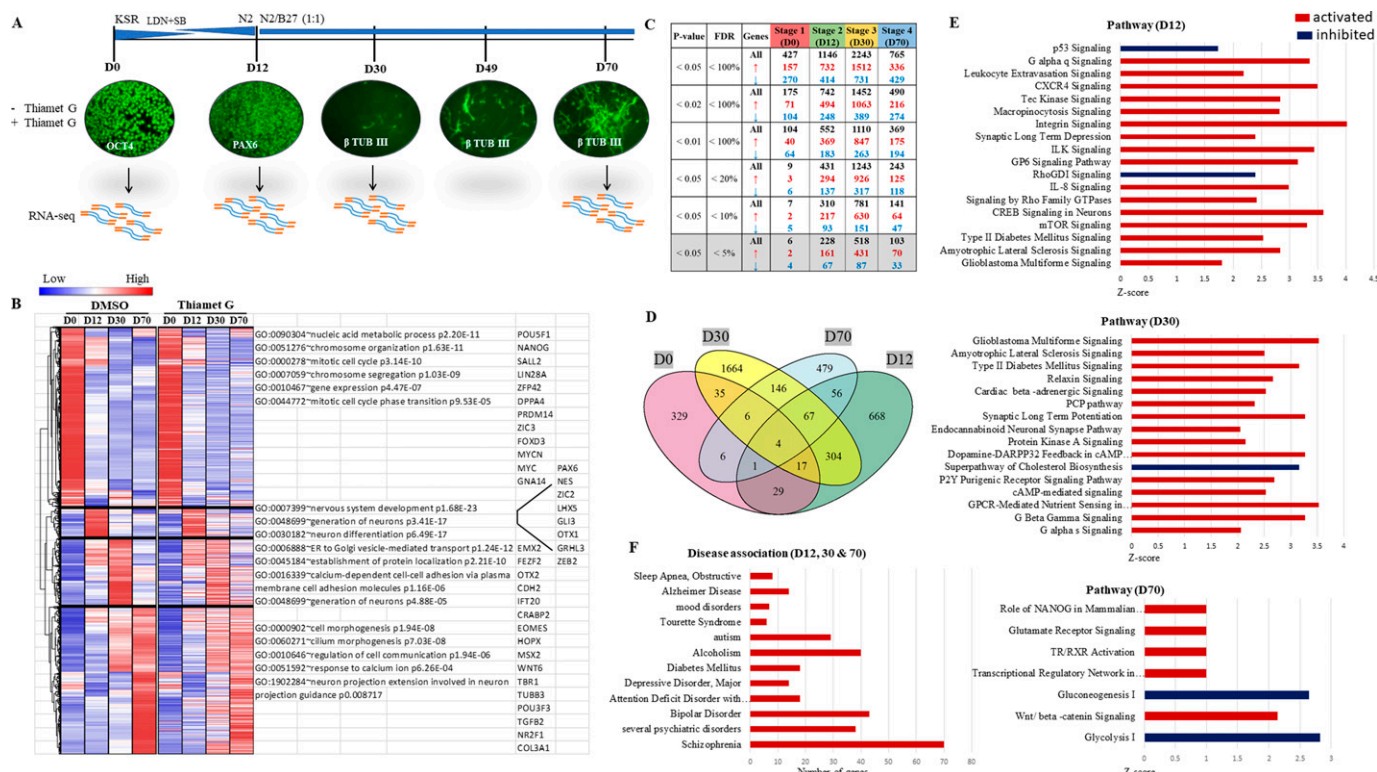

**Figure 1. Effect of elevated O-GlcNAcylation on transcriptome during embryonic cortical neurogenesis.**
**(A)** Human embryonic stem cells (H9) were differentiated into cortical neurons using dual SMAD inhibition protocol in DMEM/F12 media with knockout serum replacement (KSR) containing LDN-193189 (BMP signalling inhibitor) and SB-431542 (TGF-β Receptor Kinase inhibitor) and treated with ThiametG or DMSO as control. The cells were immunostained for stage-specific markers, OCT4 (pluripotency), PAX6 (neural stem cell), and β-TUBULIN III (early-born neuron), collected on days 0, 12, 30, and 70 of differentiation and processed for RNA-Seq (two biological replicates). **(B)** Hierarchical clustering analysis and corresponding heat map of differentially expressed genes in control and ThiametG (TMG) treated cells from four stages of neural differentiation. Hierarchical clustering was performed using Euclidean distance to visualize the expression of genes across groups that were significant in at least one inter-stage comparison with false discovery rate < 5%. **(C)** Number of genes up/down-regulated in TMG-treated cells compared to control with different *P*-values and false discovery rate. **(D)** Venn diagram of the number of genes that are significantly affected (*P*-value ≤ 0.5) due to increased *O*-GlcNAc levels from four-stages of neural differentiation. **(E)** Ingenuity pathway analysis was performed on the list of genes significantly affected (*P*-value < 0.5) in TMG-treated cells compared to control on D12, D30, and D70 of neural differentiation. The red colour indicates activated pathways, and the blue colour indicates inhibited pathways. **(F)** GAD disease association on the list of genes affected significantly (≥2-fold with *P*-value ≤ 0.5) in TMG-treated cells compared with controls on D12, D30, and D70 of neural differentiation.

studies (Parween et al, 2017; Ardah et al, 2018). This method involves dual SMAD inhibition to induce NSC fate and results in the production of excitatory glutamatergic projection neurons in a stage-specific manner, which precisely mimics in vivo human embryonic cortical neurogenesis in time and space (van de Leemput et al, 2014). Using this method, H9 hES cells were differentiated for a total of 70 d in the presence of a potent and specific inhibitor of OGA enzyme, ThiametG (TMG) or solvent (DMSO), as control. Cells were collected from four stages, day 0 (D0), day 12 (D12), day 30 (D30) and day 70 (D70) of neural induction and differentiation (Fig 1A), and were processed for RNA isolation and library preparation for RNA-seq. The libraries were subjected to RNA sequencing to identify the genes and long non-coding RNAs with altered expression due to elevated levels of *O*-GlcNAc during cortical neurogenesis. Hierarchical clustering was performed on 10,097 genes, which have shown altered expression in at least one stage out of four-stages analyzed in both control and TMG-treated cells (Fig 1B). Four major gene clusters emerged from the analysis, cluster-1 (4,262 genes), cluster-2 (718 genes), cluster-3 (1,608 genes), and cluster-4 (3,509 genes).

The cluster-1 genes were highly expressed in undifferentiated cells (Stage 1, D0 of neural induction), and their expression reduced after neural differentiation in all other three stages (D12, D30, and D70). The genes in this cluster included major pluripotency related genes such as *POU5F1*(*OCT4*), *NANOG*, *LIN28*, *MYC*, and *SALL2*, among others. Gene ontology (GO) analysis of these genes resulted in GO terms such as mitotic cell cycle, chromosome organization, nucleic acid metabolic process, etc., suggesting cell proliferation as the principal function as expected. No significant differences were noted between cluster-1 genes in control, and TMG-treated cells. Cluster-2 (D12) over-expressed genes included those of neural fate such as *PAX6*, *NES*, *LHX5*, and *OTX1*, among others. GO analysis for this cluster included terms such as nervous system development, generation of neurons and neuron differentiation, suggesting acquisition of neural fate of the ESCs. Subtle differences were noted between the control and TMG-treated cells for this cluster, indicating gene expression differences due to increased *O*-GlcNAcylation. Genes of clusters-3 and 4 included several known regulators of neuronal fate such as *EMX2*, *OTX2*, *FEZF2*, *CDH2*, and *IFT2* that were

enriched on D30, whereas early-born neuron-specific genes such as *EOMES*, *HOPX*, *TBR1*, *POU3F3* were enriched on D70, suggesting expression of genes in a stage-specific manner. GO terms for cluster-3 and4 included the generation of neurons, establishment of protein localization etc., on D30. GO-terms for D70 had processes such as neuron projection extension involved in neuron projection guidance, regulation of cell communication and response to calcium ions, suggesting stage-specific gene expression profile. Specific differences were noted between control, and TMG-treated cells where many genes from cluster-3 showed reduced expression on D30 whereas genes of cluster-4 showed increased expression of several genes on D30 which decreased in their expression on D70. This suggests an early expression of many genes on D30, which are normally expressed at a later stage (D70) (van de Leemput et al, 2014). This correlates with early and increased expression of several key neuronal differentiation genes in the presence of increased *O*-GlcNAcylation as noted in our previous study and phenotype showing early and increased production of neurons on D22 and D30 of neural differentiation (Parween et al, 2017) (Fig 1B).

Next, we used DESeq2 tool to identify gene expression differences between control and TMG-treated cells for all four stages. The overall number of differentially expressed genes increased progressively from D0 until D30 and then decreased from D30 to D70, suggesting a window of vulnerability for *O*-GlcNAc elevation. We also noticed that the number of genes up-regulated due to TMG treatment was greater by twofold or more than the number of genes that were down-regulated at all four stages of differentiation, suggesting that elevation of *O*-GlcNAc, in general, may be associated with transcriptional activation (Fig 1C). A large number of genes also showed overlap between D12 and D30 (*P* < 0.5; hypergeometric test) and D30–D70 (*P* < 0.5) (Fig 1D). Given that not many genes are differentially expressed at D0, there were not many overlaps between stage 1 and other stages. Furthermore, ingenuity pathway analysis was performed on the list of genes significantly altered in their expression (*P* < 0.5) due to TMG treatment on all four stages of differentiation (Supplemental Data 1–4). On D12, we found pathways for P53 and RhoGDI signalling were down-regulated, whereas signalling pathways for mTOR, CREB, Rho family GTPases and G α q, among others, were up-regulated. In addition, signalling for glioblastoma multiforme, type 2 diabetes, and amyotrophic lateral sclerosis were also up-regulated on D12 and D30, suggesting the association of elevated *O*-GlcNAcylation with pathological phenotype. Other signalling pathways up-regulated on D30 included G α s and G β-γ signalling, GPCR-mediated nutrient sensing, cAMP and protein kinase A signalling, among other pathways. In contrast, the super pathway of cholesterol biosynthesis was down-regulated on D30. Interestingly, we also noticed up-regulation of the planar cell polarity (PCP) signalling pathway on D30, which is associated with various processes involving cellular asymmetry in organisms ranging from worms to flies to vertebrates (Vladar et al, 2009). The major signalling pathways up-regulated in TMG-treated cells on D70 of neural differentiation included glutamate receptor signalling, Wnt-β catenin signalling, TR/RXR activation and transcription regulatory network in embryonic stem cells, whereas two major metabolic pathways, glycolysis and gluconeogenesis, were down-regulated (Fig 1E).

Next, we performed disease association analysis on the list of all genes up-/down-regulated significantly (more than twofold with *P* < 0.5) on three stages, D12, D30, and D70 of neural differentiation in TMG. Interestingly, we found major neurodevelopmental disease associations, including schizophrenia, bipolar disorder, alcoholism, and autism, which showed the most significant number of genes from GAD disease categories (Fig 1F).

These results suggest that elevated *O*-GlcNAc levels during neural differentiation significantly affect the expression of genes belonging to key signalling pathways, resulting in neuropathologies as observed through disease association analysis.

## Increased *O*-GlcNAcylation affects the expression of several key TFs of cortical neurogenesis by affecting promoter bivalency

A number of basic helix loop helix TFs are expressed at specific stages of neurodevelopment to precisely control the process of neurogenesis (Dennis et al, 2019). We noticed that the expression of several of these TFs is altered in TMG-treated cells. Ingenuity pathway analysis revealed TF enrichment among genes up-/down-regulated significantly (more than twofold with *P* < 0.5) at all four stages, D0, D12, D30, and D70 of neural differentiation in TMG-treated cells compared with control (Supplemental Data 1–4 and Fig 2A). We have also noticed a pattern in which the expression of these TFs change during different stages. The TFs involved in proliferation and maintenance of NSC population during neurogenesis including *PAX6*, *EMX2*, *SOX2*, *SOX3*, *FEZF1/2*, and notch signalling regulators, *HES*, *DLL* and other *NOTCH* genes (Lee et al, 2014) showed decreased expression in TMG-treated cells. In contrast, genes needed for neuronal differentiation, including *NEUROG1/2*, *NEUROD2/4/6*, *EOMES*, *HOPX*, *TBR1*, and several other neurogenic TFs, showed an increased expression during neuronal differentiation in TMG-treated cells with the highest up-regulation on D30 (Fig 2A). We also found that the transcription of pluripotency genes *MYC*, *OCT4*, *KLF4*, and other genes such as *KLF2*, *MYCL*, and *MYCN* is elevated in TMG in the later stages (D30 and D70) of neural induction, suggesting the possibility for neoplastic transformation. Furthermore, many of these up-regulated TF genes had their downstream gene targets up-regulated as well in TMG treatment, including known neuronal markers, *NHLH1*, *TAGLN3*, *CHGA*, *CNTN2*, and *STMN2* (Ware et al, 2016) (Supplemental Data 1–4). Because the expression of these lineage-specific TFs is tightly controlled epigenetically, we postulated the possibility of a common epigenetic mechanism involved in regulating these TFs due to elevated *O*-GlcNAc levels. To find TF/epigenetic regulator/s, which might be a common regulator of these genes in response to elevated *O*-GlcNAc levels, we used the ENCODE ChIP-Seq Significance Tool. Interestingly we found EZH2 to be enriched at significant levels at most of the genes (>1/3) affected by more than twofold (*P* ≤ 0.05) (Fig S1A). Moreover, we specifically used a list of TF genes affected due to TMG treatment to identify a common chromatin regulator targeting these TFs and again found EZH2 as the most significantly enriched protein at these genes (Fig S1B).

The role of EZH2 in the process of developmental neurogenesis is well established. Notch signalling effectors (Hes1, Hes5) along with Ezh2 are needed for the repression of proneural genes and

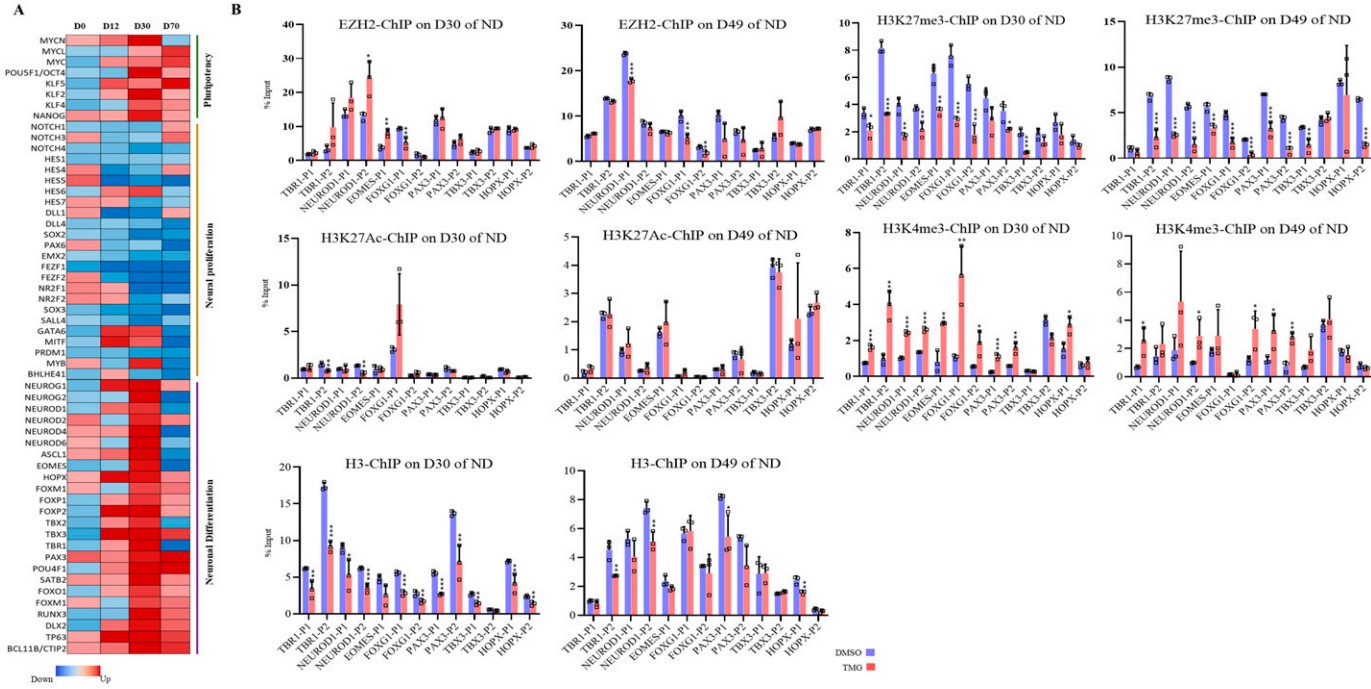

**Figure 2.  Elevated *O*-GlcNAc levels de-repress neural transcription factor genes known to be regulated by EZH2.**

**(A)** Heat map of select transcription factor genes identified by Ingenuity Pathway Analysis from the list of significantly affected genes due to ThiametG treatment from all four stages of neural differentiation. **(B)** Chromatin immunoprecipitation assay was performed to analyze the enrichment of EZH2, H3K27me3, H3K27Ac, H3K4me3, and H3 at the promoters (two sites, around −1,000 bp of transcription start site) of *TBR1*, *NEUROD1*, *EOMES*, *FOXG1*, *FOXP2*, *PAX3*, *TBX3*, and *HOPX* using specific antibodies as indicated in ThiametG-treated and DMSO control samples (days 30 and 49 of cortical differentiation). Data represent the mean of three biological replicates ± SD. Statistical analyses (unpaired *t* test) were made between control and treated samples using GraphPad Prism Software. Asterisks represent differences being significant (*$P < 0.05$, **$P < 0.01$, ***$P < 0.001$).

neuronal differentiation (Corley & Kroll, 2015). Therefore, reduced expression of notch signalling genes due to TMG as observed in RNA-seq analysis may lead to early de-repression of neurogenic TFs by possibly affecting H3K27me3 levels through EZH2. To test this, we performed chromatin immunoprecipitation (ChIP) assays for EZH2, total H3, H3K27me3, H3K4me3, and H3K27Ac on D30 and D49 of neural differentiation. We then performed real-time PCR using the ChIP'd DNA to analyze the levels of these proteins on the promoters of the selected neurogenic TFs that were up-regulated in TMG treatment (Fig 2A) on D30 and D49 of neural differentiation. EZH2 levels showed little change upon TMG treatment at the seven genes analyzed, with significant effects seen only at *NEUROD1*, *EOMES*, and *FOXG1* (Fig 2B). In contrast, the levels of H3K27me3 were significantly reduced (an average of twofold) at most of the gene promoters analyzed in TMG-treated cells compared with controls, whereas H3K4me3 levels were significantly increased (an average of twofold) at both D30 and D49 stages for these promoters. However, reduced levels of H3K27me3 were not accompanied by significant increase in H2K27Ac levels (Fig 2B). In addition, the levels of H3 were slightly lower at many of these promoters in the TMG-treated cells, consistent with nucleosome depletion at the promoters of active genes (Ozsolak et al, 2007). This depletion was substantially less than that seen for H3K27me3, so that the latter effect could not be accounted for decreased levels of H3. Thus, the levels of H3K27me3 were reduced and H3K4me3 increased on these gene promoters in TMG, despite little change in the recruitment of EZH2.

## Epigenetic dysregulations due to sustained *O*-GlcNAc levels during embryonic neurogenesis

Previous studies have shown that the stability and catalytic activity of EZH2 are affected through its posttranslational modification, including acetylation, phosphorylation, and methylation (Wan et al, 2015, 2018; Zeng et al, 2019). In addition, EZH2 is also found to be *O*-GlcNAcylated (Pereira et al, 2010; Chu et al, 2014; Lo et al, 2018). However, whether EZH2 is *O*-GlcNAcylated during embryonic neurogenesis and the function of *O*-GlcNAcylated EZH2 during neurogenesis is not known. To address this, we examined *O*-GlcNAcylation of EZH2 by co-immunoprecipitation coupled with Western blotting. We first immunoprecipitated EZH2 using an anti-EZH2 antibody from cells at D0, D12, D30, and D49 stages of neural differentiation, followed by Western blotting using an anti-*O*-GlcNAc antibody. We found that EZH2 is *O*-GlcNAcylated in undifferentiated hESCs and at all three stages of neural differentiation analyzed. Moreover, we did see a subtle increase in EZH2 *O*-GlcNAcylation in TMG-treated cells at D0 (1.9-folds) compared with control (Fig 3A and B). It is reported that *O*-GlcNAc on serine/threonine amino acid residues can compete with the phosphorylation of the same amino acids (Butkinaree et al, 2010). To test this, we performed immunoblotting of IP'd EZH2 using anti-serine/threonine phospho-antibody. Similar to *O*-GlcNAcylation, EZH2 was phosphorylated in undifferentiated hESCs and at all stages of neural differentiation; however, no major difference of EZH2

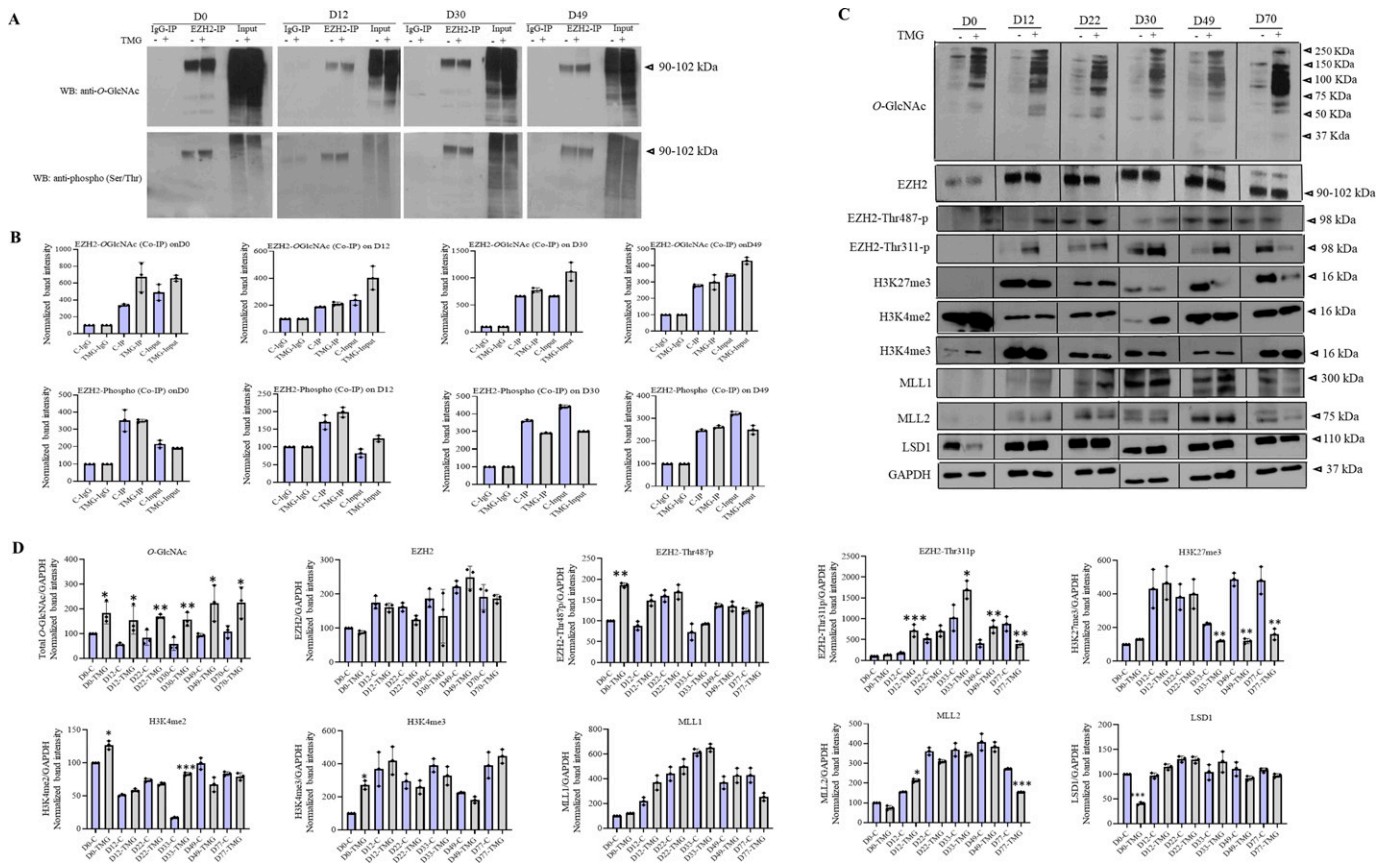

**Figure 3. Effect of elevated *O*-GlcNAc levels on EZH2 and histone modifications.**
**(A)** Co-immunoprecipitation was performed on cell lysate from ThiametG-treated and DMSO control samples from D0, D12, D30, and D49 of cortical differentiation using anti-EZH2 antibody followed by Western blotting of IP'd samples using anti-*O*-GlcNAc and anti-phospho(Ser/Thr) antibodies. IP with IgG was used as control. Input is the total cell lysate. **(B)** Densitometric quantitation of Western blots from panel (B). Data shown are mean ± SD of three biological replicates. **(C)** Western blotting was performed for *O*-GlcNAc, EZH2, EZH2-Thr487p, EZH2-Thr311p, H3K27me3, H3K4me2, H3K4me3, MLL1, MLL2, LSD1, and GAPDH on ThiametG-treated and DMSO control samples from D0, D12, D22, D30, D49, and D70 of cortical differentiation of H9 cells. The expression of GAPDH was used as the loading control. **(D)** Densitometric quantitation of Western blots from panel (C). The values are expressed as a percentage relative to the values for D0-DMSO samples set as 100%. Data represent the mean of three biological replicates ± SD. Statistical analyses (unpaired *t* test) were made between control and treated samples using GraphPad Prism Software. Asterisks represent differences being significant (*$P < 0.05$, **$P < 0.01$, ***$P < 0.001$).

phosphorylation was observed between control and TMG-treated cells except a very subtle decrease (1.24-fold) on D30 in TMG (Fig 3A and B).

To test whether EZH2 *O*-GlcNAcylation/phosphorylation may have affected its expression, we performed Western blotting for EZH2 on control and TMG-treated cells from D0, D12, D22, D30, D49, and D70 of neural differentiation. Using anti-*O*-GlcNAc antibody, we confirmed that the levels of total *O*-GlcNAc were high in all these stages in TMG-treated cells (Fig 3C and D). However, no significant difference was observed in the levels of EZH2 in TMG-treated cells compared with control cells at any of these stages of differentiation. However, these results did not rule out the possibility of changes in a specific modification site on EZH2 due to elevated *O*-GlcNAc levels. Therefore, we analyzed the level of EZH2 phosphorylation at two specific threonine (Thr) residues, Thr311 and Thr487. These EZH2 phosphorylations are reported to inhibit the H3K27 methyltransferase catalytic activity of EZH2 (Wei et al, 2011; Wan et al, 2018). Interestingly, we found EZH2-Thr487p to be higher in TMG on D0 but did not change at any other stage of differentiation. In

contrast, EZH2-Thr311p was highly up-regulated at all stages of neural differentiation in TMG-treated cells starting from D12 except on D70 where it had an opposite effect (Fig 3C and D). We also analyzed global H3K27me3, H3K4me3, and H3K4me2 and H3 levels and found H3K37me3 to be reduced in TMG on D30, D49, and D70 similar to what was reported previously due to EZH2-Thr311p (Wan et al, 2018). The levels of H3K4me2 was significantly elevated on D30 in TMG-treated cells, whereas no significant change in the levels of H3K4me3 was found at any stage of neural differentiation except increased levels on D0 in TMG treatment (Fig 3C and D). We also analyzed the expressions of H3K4 methyltransferases, MLL1 and MLL2 and H3K4 demethylase LSD1. No consistent change was observed between control and TMG-treated cells for these histone modifying enzymes, except the levels of MLL2 were slightly higher on D12 and reduced on D70, whereas LSD1 reduced on D0 in TMG-treated cells.

In summary, these results led us to conclude that many of the up-regulated genes due to elevated *O*-GlcNAc levels are key neurogenic TFs and they showed reduced H3K27me3 and increased

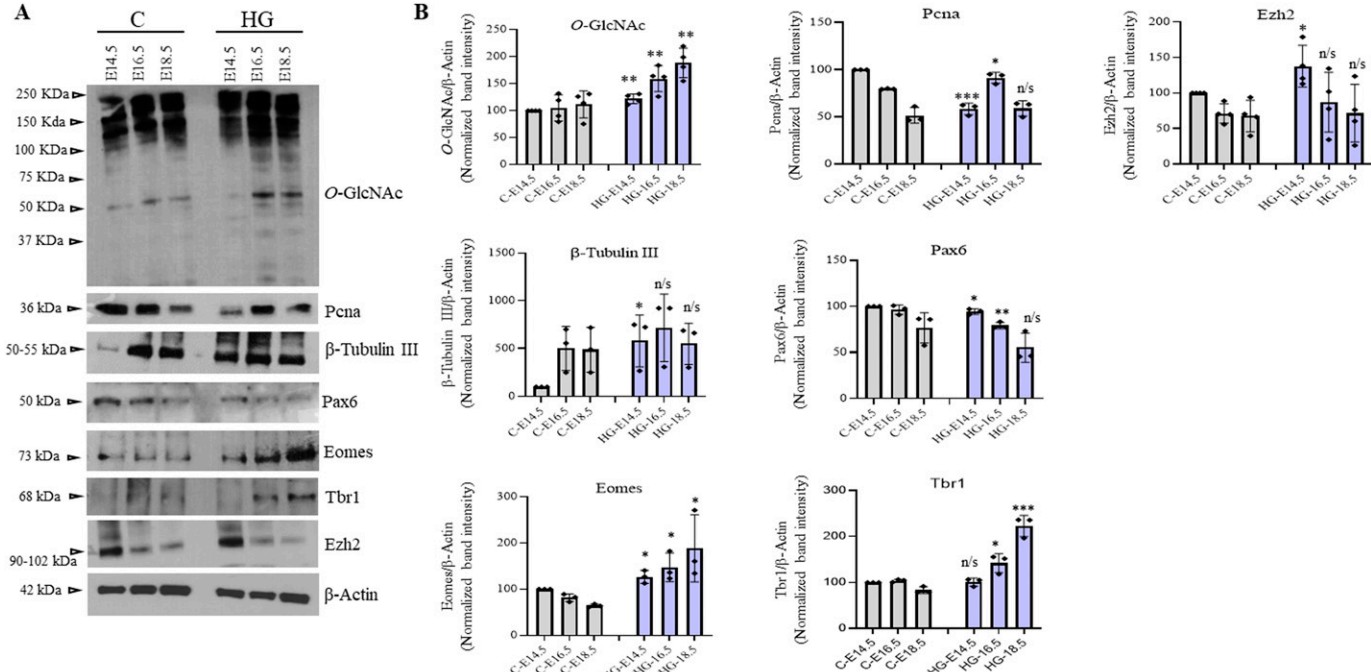

**Figure 4.  Effect of maternal hyperglycemia on the expression of markers of embryonic neurodevelopment.**
**(A)** Protein expression analysis by Western blotting was performed on embryo brain cortices from E14.5, E16.5, and E18.5 of hyperglycemia and control rats using anti-*O*-GlcNAc, PCNA, *β*-Tubulin III, Pax6, Eomes, Tbr1, and Ezh2 antibodies. The expression of *β*-Actin was used as the loading control. The results are representative of three independent biological replicates. **(A, B)** Densitometric quantitation of Western blots from panel (A). Data shown are mean ± SD of three biological replicates. Statistical analyses (unpaired *t* test) were made between control and hyperglycemia samples for each developmental stage using GraphPad Prism Software. Asterisks represent differences being significant (*P < 0.05, **P < 0.01, ***P < 0.001).

H3K4me3 on their promoters in TMG treatment. Although the binding of EZH2 on these promoters was not changed, increased EZH2-Thr311 phosphorylation due to TMG treatment could have led to the reduced H3K27 methylation levels observed. The increase in EZH2-Thr311p is evidently not due to competition with *O*-GlcNAc at the same residue; further studies will be needed to ascertain the mechanism behind this increase in phosphorylation.

### Effect of maternal hyperglycemia on protein *O*-GlcNAcylation and embryonic cortical neurogenesis

Protein *O*-GlcNAcylation is considered a major nutrient sensor and contributes to the pathology associated with hyperglycemia (Ma & Hart, 2013; Vasconcelos-Dos-Santos et al, 2018). Epidemiological and animal studies have linked hyperglycemia during pregnancy with adverse neurodevelopmental outcomes in the offspring (Kong et al, 2020; Chen et al, 2021). A recent study using a mouse model of maternal hyperglycemia has reported defects in cortical layer formation with concomitant early neuronal differentiation (Ji et al, 2019). We have also observed early and increased cortical neurogenesis in our in vitro hESCs' neuronal differentiation due to sustained *O*-GlcNAc levels (Parween et al, 2017), and it is known that hyperglycemia increases global *O*-GlcNAc levels inside the cells. So we reasoned that increased *O*-GlcNAc levels due to maternal hyperglycemia could play a role in the defective cortical differentiation as observed in these previous studies. We used a rat model of pre-gestational maternal hyperglycemia to

test the role of increased *O*-GlcNAcylation in embryonic neurogenesis. Female Wistar rats were treated with streptozotocin before mating to induce hyperglycemia during the early stages of pregnancy. Glucose levels were measured to confirm hyperglycemia (Table S1). The embryos of both control and hyperglycemic animals were collected at E14.5, E16.5, and E18.5, and the frontal lobe was dissected out under a dissection microscope. Western blotting was performed to check the effect of hyperglycemia on global protein *O*-GlcNAc levels in the developing embryo brains of hyperglycemic rats and was compared with controls. We found a modest but significant increase in the levels of *O*-GlcNAc on all three stages, E14.5, E16.5, and E18.5, in hyperglycemic embryo brains compared with controls (Fig 4A and B). Next, we checked the expression of cell proliferation marker PCNA and neuronal marker *β*-tubulin III. As reported previously in mice the expression of PCNA was low in hyperglycemic embryos at E14.5 but was slightly elevated at E16.5, whereas *β*-tubulin III was significantly up-regulated at E14.5 (Ji et al, 2019). We also checked the expression of NSC proliferation marker, Pax6 and NSC differentiation genes, Eomes and Tbr1, which had shown increased expression due to elevated *O*-GlcNAc levels in our hESCs' neuronal differentiation experiment (Parween et al, 2017). Indeed, the expression of Pax6 was low in hyperglycemic embryos at E14.5 and E16.5, whereas the expression of both Eomes and Tbr1 were up-regulated (Fig 4A and B), similar to what we have observed under high *O*-GlcNAc conditions.

These results confirm that maternal hyperglycemia reduces stem cell proliferation and increases neuronal differentiation during

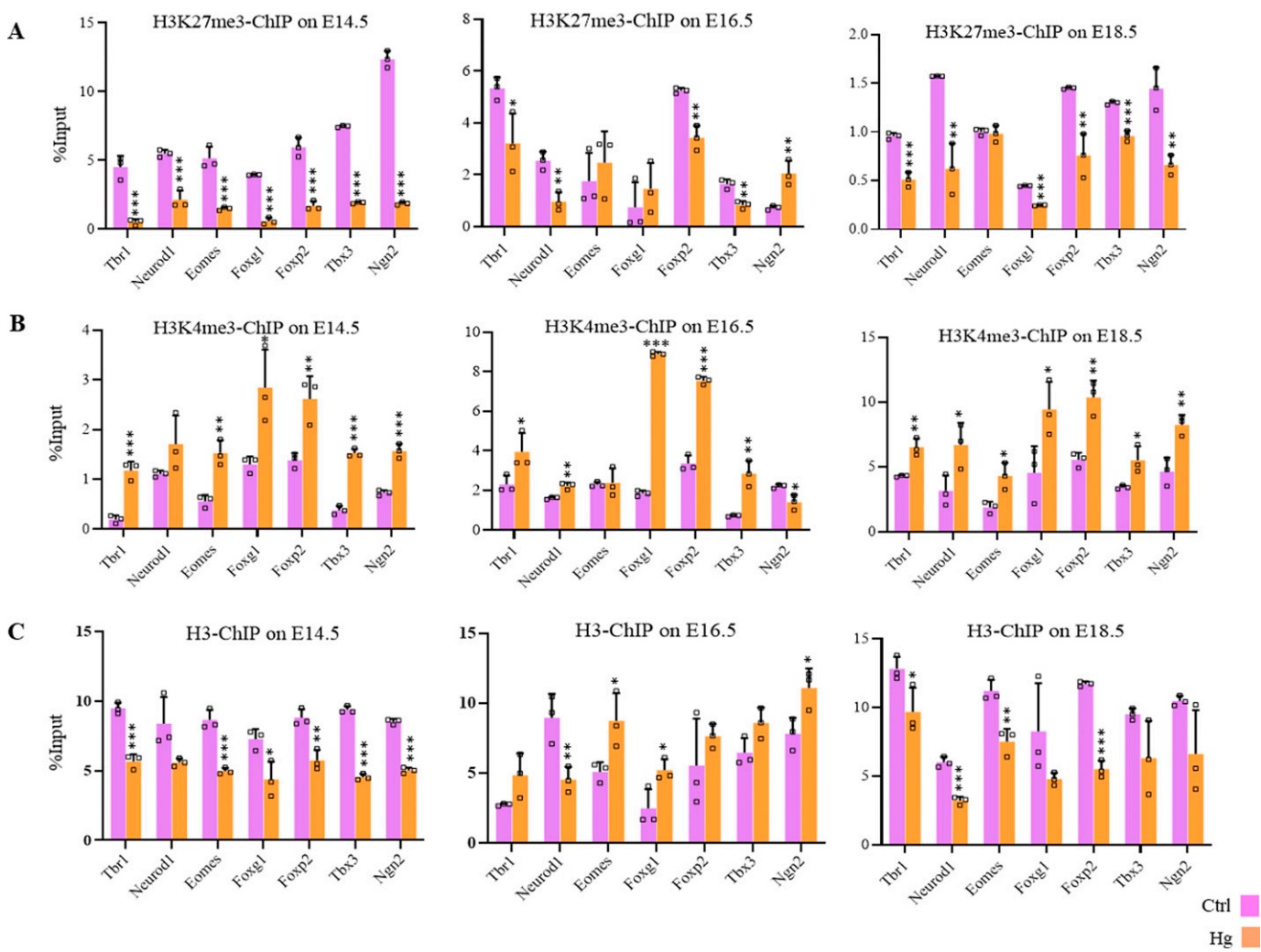

**Figure 5. Effect of maternal hyperglycemia on promoter histone methylation during embryonic neurodevelopment.**
**(A, B, C)** Embryo brain cortices from E14.5, E16.5, and E18.5 of hyperglycemic (Hg) and control (Ctrl) rats were used to perform chromatin immunoprecipitation assay for H3, H3K4me3 and H3K27me3 using chromatin immunoprecipitation grade antibodies. ChIP'd DNA was analyzed by real-time qPCR for the promoter regions of *Tbr1*, *Neurod1*, *Eomes*, *Foxg1*, *Foxp2*, *Tbx3*, and *Ngn2*. Data represent the mean of three biological replicates ± SD. Statistical analyses (unpaired *t* test) were made between control (Ctrl) and hyperglycemia (Hg) samples using GraphPad Prism Software. Asterisks represent differences being significant (*$P < 0.05$, **$P < 0.01$, ***$P < 0.001$).

development in the offspring, as previously reported (Ji et al, 2019). Importantly, these results also suggest that increased protein *O*-GlcNAc levels could be responsible for the defective corticogenesis observed due to maternal hyperglycemia in the embryos.

### Maternal hyperglycemia leads to epigenetic dysregulations in the developing embryo brains

To test whether *O*-GlcNAc–mediated epigenetic dysregulations also occur due to maternal hyperglycemia, we performed ChIP analysis of H3, H3K27me3, and H3K4me3 on embryo brain cortices from normal and hyperglycemic embryos from three developmental stages, E14.5, E16.5, and E18.5 described above. ChIP'd DNA was PCR amplified for promoter regions of seven neurogenic TF genes, *Tbr1*, *Neurod1*, *Eomes*, *Foxg1*, *Foxp2*, *Tbx3*, and *Ngn2*. Similar to what was observed due to TMG treatment in hESCs' neuronal differentiation

experiment, the levels of H3K4me3 were significantly higher (an average of 2.3-fold with $P ≤ 0.5$), and H3K27me3 were reduced (an average of 4.5-fold with $P ≤ 0.5$) on all of these gene promoters in E14.5 brain cortices in hyperglycemia. We also noticed consistently lower H3 occupancy at this stage in hyperglycemic conditions (Fig 5A–C). At E16.5 and E18.5, H3K4me3 remained high significantly in hyperglycemia except for *Eomes* (no change) and *Ngn2* (reduced). Whereas, the levels of H3K27me3 remained lower in hyperglycemia at these two stages except for *Eomes* and *Foxg1* (no change) and *Ngn2* (showed up-regulation) at E16.5 and no change was observed for *Eomes* on E18.5. The total H3 also showed a subtle decrease at E18.5 consistent with what was observed at E14.5; however, at E16.5, an opposite trend was observed where H3 levels showed a modest increase in hyperglycemia on all promoters analyzed except *Neurod1* (Fig 5A–C).

These results suggest that maternal hyperglycemia affects promoter bivalency on key neurogenic TF genes in the developing

embryo brains. This also raises the possibility that increased *O*-GlcNAcylation due to maternal hyperglycemia could be responsible for de-repression of these genes due to an imbalance of bivalent marks on promoters.

### Maternal hyperglycemia and *O*-GlcNAc elevation affect H2B *O*-GlcNAcylation and H2B mono-ubiquitination and up-regulate Pol II Serine phosphorylation

Previous studies have shown histone H2B to be *O*-GlcNAcylated at serine 112, promoting H2B mono-ubiquitination at lysine 120 (Fujiki et al, 2011; Xu et al, 2014). Moreover, H2BK120 mono-ubiquitination stimulates H3K4me3 in humans and other organisms (Zhu et al, 2005; Kim et al, 2009). Other studies have reported that the role of OGT is to promote H3K4me3 in a TET2/TET3 dependent manner (Deplus et al, 2013).

We have seen that H3K4me3 levels were increased due to elevated *O*-GlcNAc levels at D30 and D49 of neural differentiation and also due to maternal hyperglycemia at the neurogenic TF gene promoters. So we asked whether the levels of H2BS112*O*-GlcNAcylation and H2BK120Ub1 are also altered on these neurogenic TF gene promoters due to elevated *O*-GlcNAc levels. To address this, we performed ChIP using anti-H2B S122-*O*GlcNAc and anti-H2BK120Ub1 antibodies on hESCs from D30 and D49 of neural differentiation. We used the ChIPed DNA and performed RT-qPCR on six TF genes (*TBR1*, *NEUROD1*, *EOMES*, *FOXG1*, *FOXP2*, and *PAX3*) which were up-regulated on D30 of neural differentiation in TMG treatment and three genes (*NR2E1*, *HES5*, and *FEZF2*) which were down-regulated. We did not find a consistent change in the occupancy of H2B S122-*O*GlcNAc and H2BK120Ub1 due to TMG on these promoters. However, a subtle reduction in H2B occupancy was seen at some of these promoters on both D30 and D49 in TMG, whereas four of the nine genes (*TBR1*, *PAX3*, *NR2E1*, and *FEZF2*) showed increased H2B occupancy at D49 in TMG. Similarly, H2B S122-*O*GlcNAc were also increased for these same genes at D49 whereas the levels of H2BK120Ub1 were elevated on all genes promoters tested except for *HES5*. However, on D30, only *NEUROD1* and *FOXP2* showed more than twofolds increase for H2BK120Ub1 in TMG. Thus, unlike H3K4me3 and H3K27me3 levels, which showed consistent changes on all promoters analyzed on both D30 and D49 in TMG, clear changes were not observed for H2B S122-*O*GlcNAc and H2BK120Ub1 (Fig 6A and B).

Previous reports have also shown that H2B mono-ubiquitin positively regulated transcription by RNA Pol II (Pavri et al, 2006; Tanny et al, 2007). Therefore, we decided to analyze by ChIP the recruitment of unmodified-Pol II (un-Pol II) and Ser5p-Pol II (associated with active transcription initiation) on eight neurogenic TF genes (*TBR1*, *EOMES*, *NEUROD1*, *FOXG1*, *FOXP2*, *PAX3*, *HOPX*, and *TBX3*) which have shown transcriptional up-regulation in TMG. ChIP performed on D30, and D49 of neural differentiation in control and TMG-treated cells showed decreased un-Pol II occupancy on all these genes on D30 and at some of the genes on D49 in TMG. In contrast, the levels of Ser5p-Pol II were consistently elevated on most of the promoters in TMG on both D30 and D49 (Fig 6C and D) (asterisks indicate differences being significant). These results suggest a general increase in active transcription of these genes due to TMG, resulting in transcriptional up-regulation observed for these genes on D30 in RNA-seq data.

It has also been reported that the levels of histone *O*-GlcNAcylation fluctuate in response to extracellular glucose (Fujiki et al, 2011). As we have already observed the global *O*-GlcNAc to be high in the embryo's brain of hyperglycemic animals compared with controls, we performed Western blotting to analyze the expression of total H2B, H2BS122*O*GlcNAc, and H2BK120Ub1 from all three developmental stages, E14.5, E16.5, and E18.5. The levels of total H2B decreased gradually from E14.5 to E18.5; however, no significant difference was observed in the levels of total H2B (except a modest reduction at E18.5) and H2BK120Ub1 between control and hyperglycemic embryos. In contrast, the levels of H2BS122*O*GlcNAc was higher in the embryos from hyperglycemic mothers at all three stages (Fig 7A and B) and were significant at E14.5 and E18.5. We then performed ChIP for total H2B, H2BS122*O*GlcNAc, and H2BK120Ub1 from all these developmental stages (E14.5, E16.5, and E18.5) from hyperglycemic and control embryos brains and performed RT-qPCR for the seven TF genes, *Tbr1*, *Neurod1*, *Eomes*, *Foxg1*, *Foxp2*, *Tbx3*, and *Ngn2*. We found that the levels of total H2B were low at all seven genes at E14.5 and E18.5; however, at E16.5, an opposite trend was observed on all genes except *Foxg1* and *Ngn2*. A general increase in the levels of H2BS122*O*GlcNAc was seen in case of hyperglycemia compared with controls at all genes except *Foxp2* at E14.5 and *Ngn2* at E16.5. On the other hand, H2BK120Ub1 levels did not change at E14.5, whereas a general increase was seen in hyperglycemic embryos for all genes at E16.5 and E18.5 except *Tbx3* on E18.5 (Fig 7C–E).

We also performed ChIP for un-Pol II and Pol II-Ser5p and analyzed the enrichment in control and hyperglycemia at the promoters of the same seven genes as above. The changes in the recruitment of un-Pol II in hyperglycemia compared with control on these genes was inconsistent for most genes except *Foxg1* which showed increased un-Pol II occupancy in hyperglycemia on all three stages, E14.5, E16.5, and E18.5 (Fig 8A). In contrast to this, the levels of Pol II-Ser5p were consistently and significantly higher in hyperglycemia for all genes analyzed except *Eomes* at E14.5, *neurod1* at E16.5, and *Ngn2* at E18.5 (Fig 8B). This is similar to what was observed in hESCs' neuronal differentiation on D30 and D49, where an increased occupancy of Pol IISer-5p was observed at many of the gene promoters analyzed (Fig 6C and D).

These results show that hyperglycemia results in global up-regulation of H2B-*O*GlcNAcylation at S112 and increased occupancy on the promoters of neurogenic TF genes analyzed even though the trend is not entirely consistent for all genes analyzed. This probably could have led to increased H2BK120Ub1 on these genes as well. However, this was not the case during hESCs' neuronal differentiation. Notably, both increased *O*-GlcNAc levels in human hESCs derived neural cells and hyperglycemia in animals led to consistently increased Pol II-Ser5p occupancy on these genes. This is also consistent with the gene up-regulation due to elevated *O*-GlcNAc levels observed in RNA-seq data.

### OGT inhibition restored maternal hyperglycemia-mediated molecular changes in the embryo brain

The above experiments showed similar epigenetic dysregulations in the developing embryo brain cortices due to maternal hyperglycemia to that observed due to elevated *O*-GlcNAc levels during

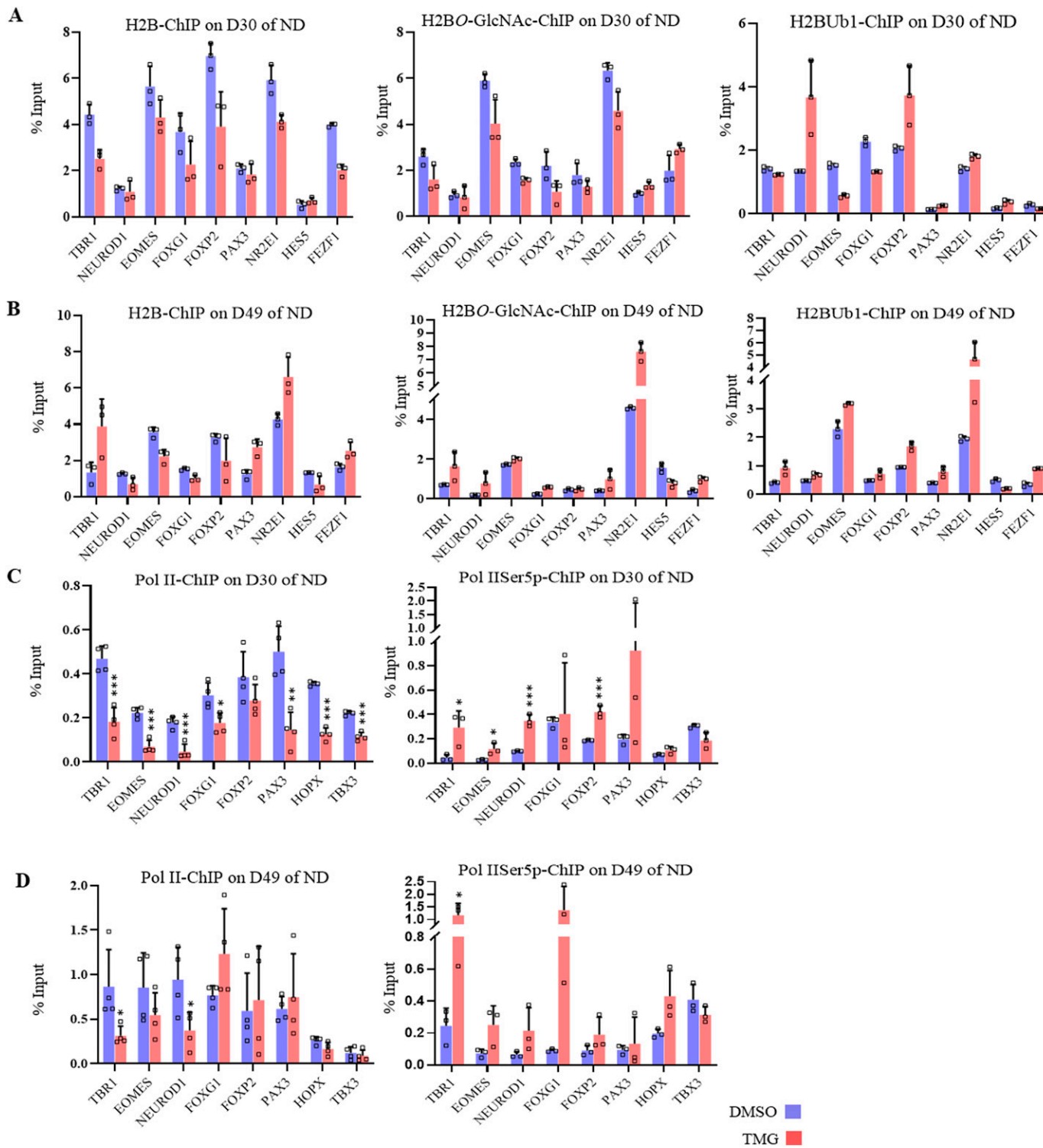

**Figure 6. Effect of high *O*-GlcNAc levels on promoter H2B *O*-GlcNAcylation/mono-ubiquitination and Pol II-Serine5/2 phosphorylation.**
**(A, B, C, D)** Chromatin immunoprecipitation assay was performed to analyze the enrichment of histone, H2B, H2BS112*O*-GlcNAc and H2BK120Ub1at the promoter (A, B) and unmodified Pol II, and Pol II-Ser5p (C, D) at 5′ of the coding region of the gene of *TBR1*, *NEUROD1*, *EOMES*, *FOXG1*, *FOXP2*, *PAX3*, *NR2E1*, *HES5*, and *FEZF1* using chromatin immunoprecipitation grade antibodies as indicated in ThiametG-treated and DMSO control samples (day 30 and 49 of cortical differentiation). Data represent the mean of three biological replicates ± SD. Statistical analyses (unpaired *t* test) were made between control and treated samples using GraphPad Prism Software. Asterisks represent differences being significant (\*P < 0.05, \*\*P < 0.01, \*\*\*P < 0.001).

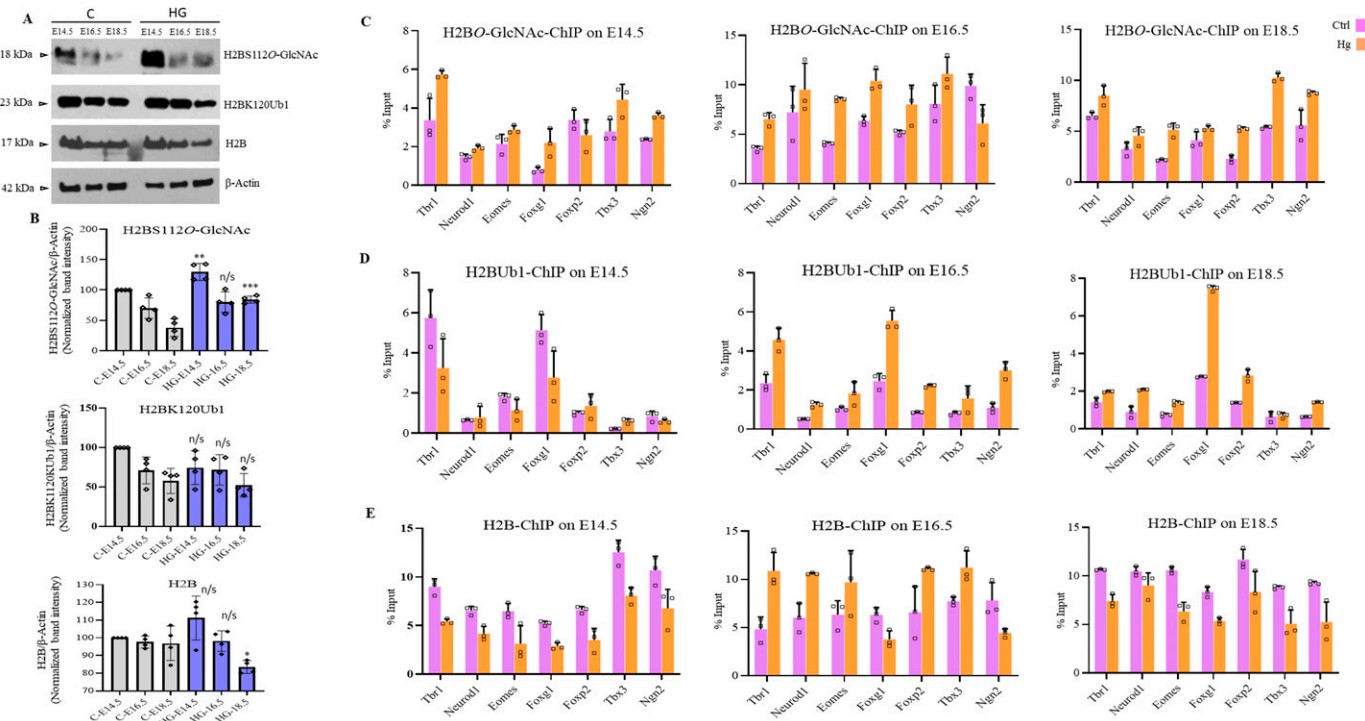

**Figure 7. Effect of maternal hyperglycemia on histone H2B *O*-GlcNAcylation/mono-ubiquitination during embryonic neurodevelopment.**
**(A)** Protein expression analysis by Western blotting was performed on embryo brain cortices from E14.5, E16.5, and E18.5 of hyperglycemia (HG) and control (C) rats using anti-H2B, H2BS112*O*-GlcNAc, and H2BK120ub1 antibodies. The expression of *β*-Actin was used as the loading control. **(B)** Densitometric quantitation of Western blots from panel (B). Data shown are mean ± SD of three biological replicates. Statistical analyses (unpaired *t* test) were made between control and hyperglycemia samples for each developmental stage using GraphPad Prism Software. Asterisks represent differences being significant (*$P < 0.05$, **$P < 0.01$, ***$P < 0.001$). **(C, D, E)** Embryo brain cortices from E14.5, E16.5, and E18.5 of hyperglycemic (Hg) and control (Ctrl) rats were used to perform chromatin immunoprecipitation assay for H2B, H2BS112*O*-GlcNAc and H2BK120ub1 using chromatin immunoprecipitation grade antibodies. ChIP'd DNA was analyzed by real-time qPCR for the promoter regions of *Tbr1*, *Neurod1*, *Eomes*, *Foxg1*, *Foxp2*, *Tbx3*, and *Ngn2*. Data shown are mean ± SD of three biological replicates.

hESCs' neuronal differentiation, especially the decreased H3K27me3 and increased H3K4me3 levels on the gene promoters analyzed. To confirm whether these maternal hyperglycemia-induced epigenetic changes were indeed due to increased *O*-GlcNAc levels, we injected pregnant hyperglycemic rats with OGT inhibitor ST045849 for 4 d (E9.5 to E13.4) and collected embryo brain cortices at E14.5. Western blotting on control (C), hyperglycemia (HG), and hyperglycemia treated with ST045849 (ST) rat embryo brains showed elevated *O*-GlcNAc levels due to HG compared with C, which was reduced to almost normal levels after treatment with ST. This result suggested that ST had successfully inhibited OGT action in the embryo brain, similar to what was reported previously (Kim et al, 2017). We next performed Western blottings for Ezh2-Thr311p, Ezh2-Thr487p, Ezh2, histone H3, H3K4me2, H3K4me3, H3K27me3, and histone modification enzymes, Mll1, Mll2, and Lsd1 using these rat embryo brain cortices (Fig 9A). Interestingly, we found the levels of Ezh2-Thr311p to be significantly elevated in HG compared with C and this was reduced to almost C levels due to OGT inhibition. This again is consistent with the increased EZH2-Thr311p observed due to TMG treatment during hESCs' neuronal differentiation (Fig 3C and D). The levels of Ezh2-Thr487p was also elevated in HG but did not reduce due to ST treatment, whereas levels of total Ezh2 did not change (Fig 9A and B). Among the histone modifications, both H3K4me2 and H3K4me3 were elevated in HG; however, only H3K4me2 levels were

reduced significantly after ST treatment. The levels of both H3K27me3 and total H3 did not change. Interestingly, the expression of Mll1 was elevated significantly in HG and drastically reduced due to OGT inhibition, whereas Mll2 and Lsd1 did not show any change in HG or treatment with ST (Fig 9A and B). We also found that the expression of neuronal TF genes, Eomes and Tbr1 and neuronal marker, *β*-Tubulin III up-regulated in HG was reduced to almost C levels due to ST treatment (Fig S2).

The striking result, the up-regulation of Ezh2-Thr311 phosphorylation in HG and its reduction due to OGT inhibition, suggests that Ezh2-Thr311 phosphorylation is dependent on *O*-GlcNAcylation. Furthermore, the sensitivity of Mll1 to HG and OGT is also remarkable as increased Mll1 levels due to high glucose could have led to increased H3K4me3 levels on the promoters of neuronal TF genes as observed in ChIP experiments.

## Discussion

Nutrient sensing and signalling pathways regulate normal embryo development by regulating cell fate decisions (Rafalski et al, 2012; Ochocki & Simon, 2013; Folmes & Terzic, 2014). Among other nutrient signalling mechanisms, protein *O*-GlcNAcylation through the action of a pair of enzymes, OGT and OGA, has emerged as a major

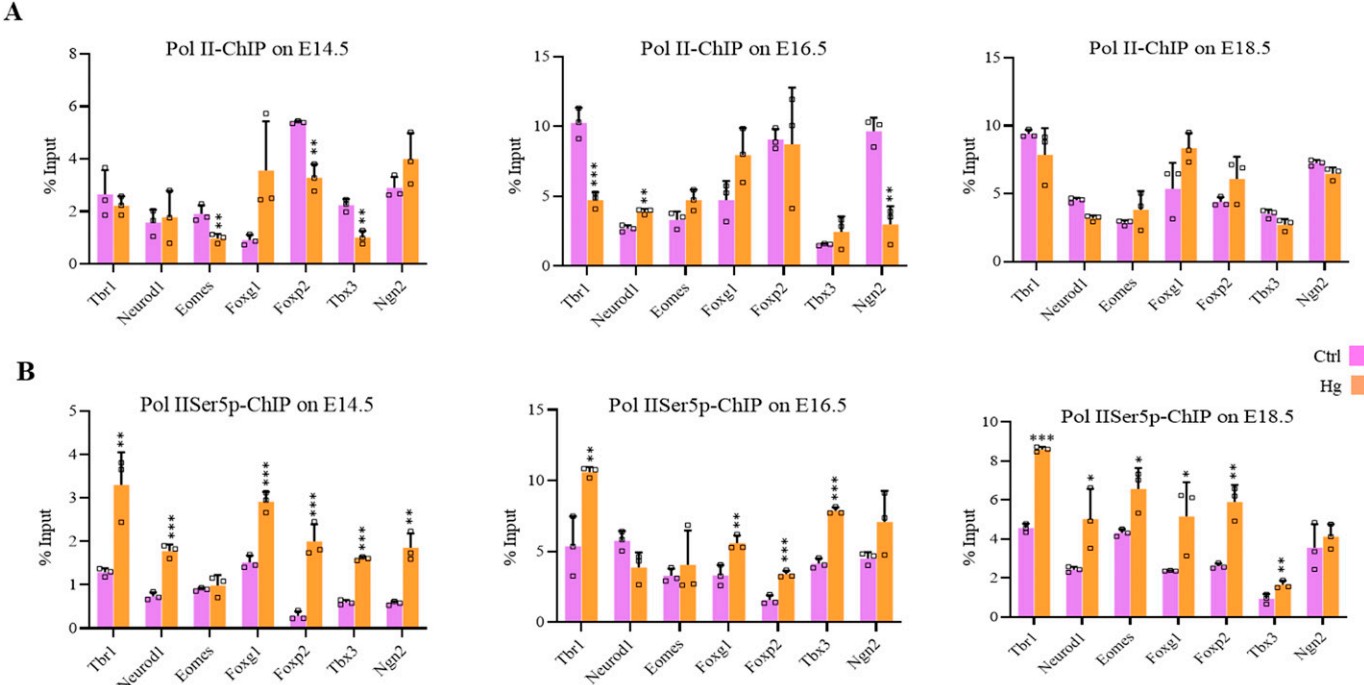

**Figure 8. Effect of maternal hyperglycemia on Pol II-Serine5/2 phosphorylation during embryonic neurodevelopment.**
**(A, B)** Embryo brain cortices from E14.5, E16.5, and E18.5 of hyperglycemic (Hg) and control (Ctrl) rats were used to perform chromatin immunoprecipitation assay for unmodified Pol II and Pol II-Ser5p using chromatin immunoprecipitation grade antibodies. ChIP'd DNA was analyzed by real-time qPCR to analyze recruitments at 5′ of the coding region of the gene of *Tbr1*, *Neurod1*, *Eomes*, *Foxg1*, *Foxp2*, *Tbx3*, and *Ngn2*. Data represent the mean of three biological replicates ± SD. Statistical analyses (unpaired *t* test) were made between control (Ctrl) and hyperglycemia (Hg) samples using GraphPad Prism Software. Asterisks represent differences being significant (*$P < 0.05$, **$P < 0.01$, ***$P < 0.001$).

regulator of nutrient signal transduction (Yang & Qian, 2017; Ong et al, 2018) and epigenetic gene regulation (Lewis & Hanover, 2014; Hardivillé & Hart, 2016). Nutrient abundance, as in conditions of hyperglycemia, is shown to alter cell fate decisions during neurodevelopment by inducing early and increased neuronal differentiation (Gao & Gao, 2007; Ji et al, 2019). Abnormal glucose levels, as in hyperglycemia, also affect global *O*-GlcNAc levels (Ma & Hart, 2013; Peterson & Hart, 2016; Masaki et al, 2020). Therefore, to understand the role of elevated protein *O*-GlcNAc levels on transcriptional/epigenetic gene expression and regulation during embryonic neurogenesis, we have used neuronal differentiation of hESCs as a model. We also extended this study to validate identified *O*-GlcNAc-mediated gene regulatory mechanisms in an in vivo rat model of maternal hyperglycemia (Fig 10). We found that increased *O*-GlcNAc levels de-repress several neurogenic TF genes, particularly on D30 of neural induction of hESCs. Most of these TF genes are reported to have bivalent histone marks, H3K27me3 and H3K4me3 (Albert et al, 2017), in the early stages of neurogenesis. Whole-genome transcriptome analysis during embryonic neurogenesis has been performed in both rodents, human fetal tissues and in vitro models of hPSCs-neurogenesis to identify gene expression profiles of specific stem cell lineages during neurogenesis (Han et al, 2009; van de Leemput et al, 2014; Fan et al, 2018). Moreover, epigenomic profiling has led to the identification of unique cell-type-specific DNA and histone modifications associated with these lineages (Yao et al, 2016; Amiri et al, 2018). During in vitro hESCs'

neurogenesis, pluripotent cells acquire neuroepithelial fate (NECs) by day 5–7, which become NSCs by day 12 (van de Leemput et al, 2014). These NSCs then differentiate into neural progenitor cells (NPCs) and give rise to early-born neurons starting around day 30–35 of neural induction. These early-born neurons are similar to deep layer neurons formed during mammalian in vivo cortico-genesis (van de Leemput et al, 2014). These NECs, NSCs, NPCs and early-born neurons temporally express several unique lineage-specific TFs. Many of these proneural and neurogenic, basic helix loop helix TFs along with specific epigenetic mechanisms has been shown to confer cell type specification during neurogenesis (Imayoshi & Kageyama, 2014; Dennis et al, 2019). Histone modifications during neurogenesis constitute a major epigenetic mechanism responsible for cell-type specification (Lomvardas & Maniatis, 2016). H3K27me3, a gene repression mark and H3K4me3, a transcriptional activation mark, are two histone modifications associated with cell fate specification. Interestingly, both H3K27me3 and H3K4me3 are identified together on promoters of many developmental genes in both hESCs and during neurogenesis (Vastenhouw et al, 2010; Vastenhouw & Schier, 2012; Voigt et al, 2013), and such promoters are termed as "bivalent." It has been proposed that bivalent genes, many of which are stage-specific TF genes, are poised to be activated or silenced during specific stages of development and thus regulate cell fate decisions (Vastenhouw & Schier, 2012). H3K27me3 is deposited by Polycomb group (PcG) complex catalyzed by histone methyltransferase, EZH2. PcG

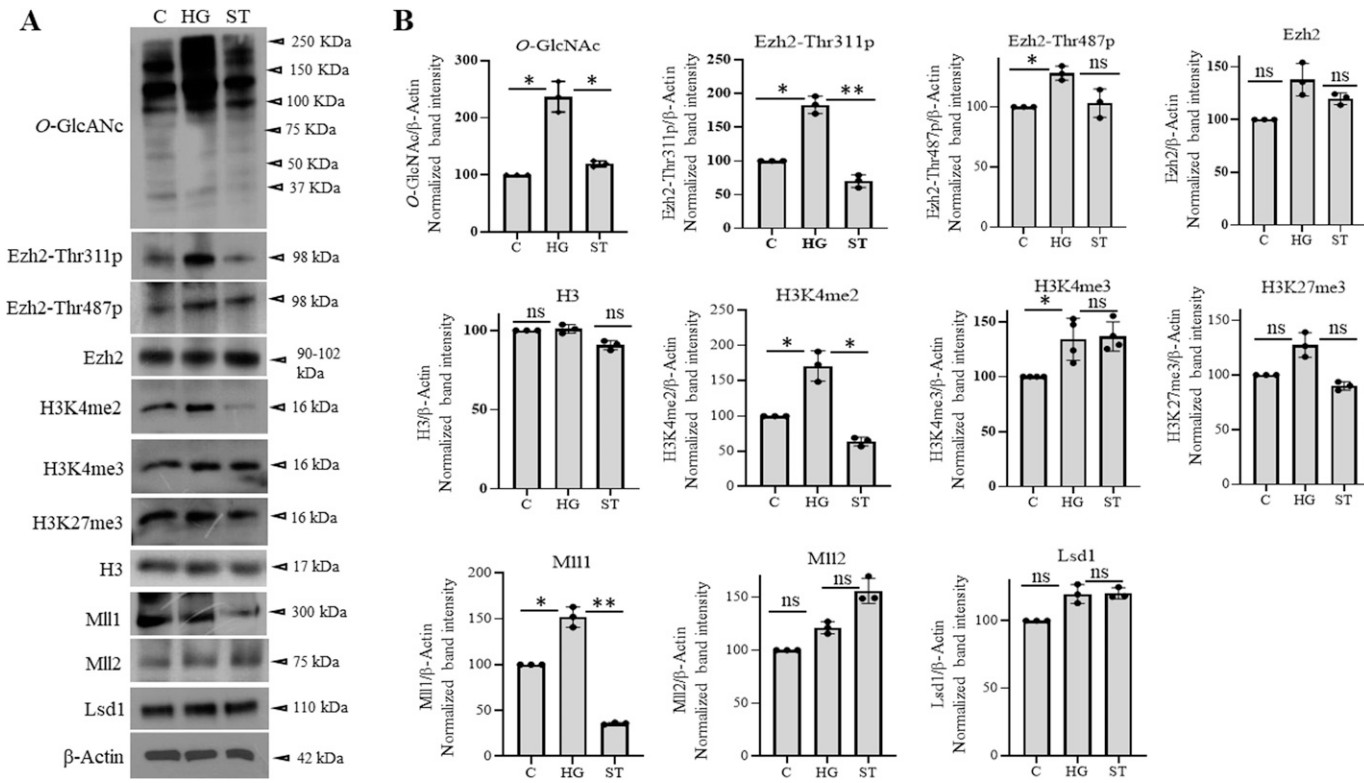

**Figure 9. The effect of *O*-GlcNAc transferase inhibition on maternal hyperglycemia-mediated molecular changes in developing embryo brains.**
**(A)** Protein expression analysis by Western blotting was performed on embryo brain cortices from E14.5 of control (C), hyperglycemic (HG), and hyperglycemic treated with *O*-GlcNAc transferase inhibitor, ST045849 (ST) rats using anti-*O*-GlcNAc, Ezh2-Thr311p, Ezh2-Thr487p, Ezh2, H3K4me2, H3K4me3, H3K27me3, H3, Mll1, Mll2, and Lsd1 antibodies. The expression of *β*-Actin was used as the loading control. The results are representative of three independent biological replicates. **(A, B)** Densitometric quantitation of Western blots from panel (A). Data shown are mean ± SD of three biological replicates. Statistical analyses (unpaired *t* test) were made between C versus HG samples and HG versus ST samples using GraphPad Prism Software. Asterisks represent differences being significant (*$P < 0.05$, **$P < 0.01$, ***$P < 0.001$).

proteins play significant roles during neurogenesis by balancing the self-renewal and differentiation of NPCs (Pereira et al, 2010; Albert et al, 2017). We found EZH2 to be enriched at many genes de-repressed due to increased *O*-GlcNAc levels, including lineage-specific neurogenic TF genes. This prompted us to investigate the role of EZH2 and promoter bivalency on some of these de-repressed TF genes. Although the levels of EZH2 were stable, we found Thr311 phosphorylation of EZH2 to be increased under *O*-GlcNAc elevation in hESCs' neuronal cell culture and in vivo in the embryo brains of hyperglycemic rats (Fig 10). As reported previously, EZH2-Thr311p suppresses its H3K27 methyltransferase activity but does not inhibit its association with histone, H3 (Wan et al, 2018). Consistent with this, we found a global reduction in H3K27me3 levels at D30–D70 of neural differentiation and reduced H3K27me3 levels on the promoters of neurogenic TF genes under high *O*-GlcNAc conditions in human cells. Similarly, H3K27me3 levels were reduced on the same promoters in embryo brain cortices due to maternal hyperglycemia. Conversely, H3K4me3 was simultaneously up-regulated at these genes. Interestingly, the protein levels of H3K4 methyltransferase, Mll1 were up-regulated due to hyperglycemia in embryo brains and treatment with OGT inhibitor restored Mll1 to normal levels, suggesting that this increased Mll1 could have resulted in H3K4me3 enrichment on the affected gene promoters.

Alternatively, histone H2BK120Ub1 may also have played a role in increased H3K4me3 levels observed in our study. Previous studies have shown a role for H2Bub1 in H3K4me3 at gene promoters (Vitaliano-Prunier et al, 2008; Kim et al, 2009) and H2B *O*-GlcNA-cylation at S122 was shown to promote H2BK120Ub1 (Fujiki et al, 2011; Xu et al, 2014). We expected increased H2BS122*O*-GlcNAc levels in TMG-treated cells during neural differentiation due to globally elevated *O*-GlcNAc levels. Surprisingly we found that the levels of H2BS122*O*-GlcNAc were not elevated on the promoters analyzed on D30 of neural differentiation. On the contrary, we found a subtle decrease of H2BS122*O*-GlcNAc in TMG, similar to that observed for un-modified H2B. However, we did find increased H2BK120Ub1 levels in TMG for some of the genes analyzed on D30. On D49, however, we did see increased H2BS122*O*-GlcNAc and H2BK120Ub1 levels in TMG for many of the genes studied. In the embryo brains of hyperglycemic rats, we noticed a more consistent increase in H2BS122*O*-GlcNAc and H2BK120Ub1 levels on the promoters ana-lyzed. This suggests that increased *O*-GlcNAc levels and hyper-glycemia may increase H2BS122*O*-GlcNAc and H2BK120Ub1 levels on the gene promoters, resulting in increased H3K4me3 levels observed on these genes. However, it is also possible that H2BS122*O*-GlcNAc and H2BK120Ub1 could be more dynamic or cell type-specific. Because our hESCs' neuronal cultures and brain

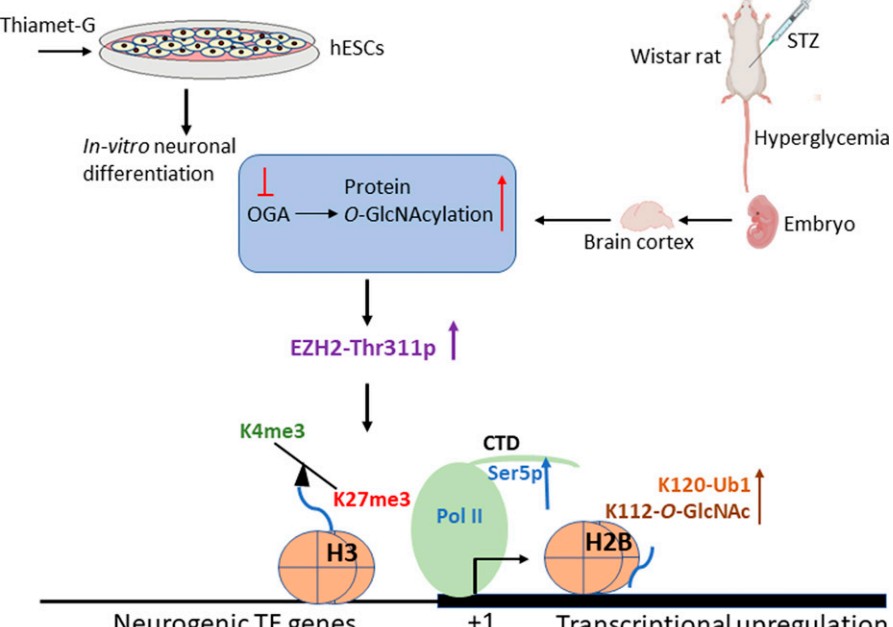

**Figure 10. Model connecting hyperglycemia and elevated *O*-GlcNAc levels to epigenetic dysregulation and gene expression changes during in vitro and in vivo embryonic neurogenesis.**
Based on our findings, pharmacological up-regulation of *O*-GlcNAc levels during human embryonic stem cells' neuronal differentiation in vitro and maternal hyperglycemia in rats which elevates global *O*-GlcNAc levels in the developing embryo brain cortex leads to increased phosphorylation of EZH2 at Thr311, resulting in decreased H3K27me3 and subsequent transcriptional up-regulation through increased Pol II Ser5-p, H3K4me3, H2BS112-*O*-GlcNAc, and H2BS120-Ub1 levels.

cortices used, both have heterogenous cell populations, it will be essential to validate these results in purer populations of cells in future studies. The role of H2BK120Ub1 in transcription elongation is recognized in model organisms, including yeast and human cells (Tanny et al, 2007). In humans, RAD6 and BRE1 homologs are suggested to function as the relevant ubiquitin-conjugating (E2) and ubiquitin ligase (E3) enzymes, respectively (Kim et al, 2009). Both RAD6 and BRE1 are recruited to the promoters and interact with elongating (hyperphosphorylated) Pol II and PAF complex (Xiao et al, 2005). We have also found Pol II-Ser5p levels were consistently elevated on all of the genes analyzed in both TMG-treated cells and in the condition of hyperglycemia. This suggests that Pol II-Ser5p could also modulate H2BK120Ub1 levels, which could then regulate H3K4 trimethylation. Indeed, it is reported that in yeast, loss of C-terminal domain (CTD) of Pol II or inactivation of Kin28, the kinase responsible for Ser5 phosphorylation of Pol II CTD, leads to dramatically reduced levels of H2BK120Ub1 (Xiao et al, 2005). Therefore, both H2BS122*O*-GlcNAc and Pol II-Ser5p could regulate H2BK120Ub1, which could then modulate H3K4methylation, ultimately affecting transcription. However, what could have caused the up-regulation of Pol II-Ser5p levels under higher *O*-GlcNAc or hyperglycemic conditions on the promoters of de-repressed genes? Previous reports suggest that Pol II CTD is *O*-GlcNAcylated (Ranuncolo et al, 2012) and human cell-free transcription assays provide evidence of *O*-GlcNAcylated Pol II on promoters (Lewis et al, 2016). Interestingly, it was reported that CTD of Pol II is *O*-GlcNAcylated on Ser5 and Ser2 and removal of *O*-GlcNAc would be needed for Pol II to enter into transcriptional initiation/elongation. Therefore, in the conditions of excess *O*-GlcNAcylation, increased Pol II *O*-GlcNAcylation could result in reduced transcriptional initiation/elongation. Arguing against this scenario, we found an increase in Pol II-Ser5p levels under high *O*-GlcNAc and hyperglycemic conditions on promoters analyzed by ChIP (Fig 10). Therefore, it will be of interest to know whether antagonism between Pol II CTD *O*-GlcNAcylation and phosphorylation is true on every promoter in vivo or is this mechanism regulated by specific environmental conditions such as stress and/or nutrient abundance/scarcity? In fact, a genome wide study carried out in *Caenorhabditis elegans* does show nutrient dependent role of *O*-GlcNAc in regulating Pol II dynamics and transcriptional regulation (Krause et al, 2018).

In summary, our work has identified specific gene regulatory mechanisms mediated through protein *O*-GlcNAcylation during embryonic neurogenesis. These include its effect on EZH2-Thr311 phosphorylation, Mll1 expression and associated histone modifications, Pol II Ser5 phosphorylation and H2B *O*-GlcNAcylation/mono-ubiquitination. These mechanisms, perturbed due to maternal hyperglycemia and elevated *O*-GlcNAc levels, ultimately result in dysregulation of gene expression. This includes early and increased expression of neurogenic TF genes, which affects cell fate decisions during embryonic neurogenesis. Such mechanisms could be responsible for adverse neurodevelopmental outcomes observed due to maternal hyperglycemia in human epidemiological studies. Therapeutic targeting of OGT could be a novel strategy to overcome the adverse effects of maternal hyperglycemia or similar metabolic perturbations.

# Materials and Methods

### hESC culture and differentiation

Commercially available hESC line, H9 (WA09), was procured from WiCell Institute Madison and used throughout this study. The cells

were cultured in feeder-free conditions on Matrigel and differentiated towards cortical neurons exactly as previously reported (Parween et al, 2017; Ardah et al, 2018). Briefly, Matrigel-coated six-well tissue culture plates (P6 plate) (Cat. no. 3516; Corning) were used to culture H9 cells in mTeSR1 medium (Cat. no. 05850; Stem Cell Technologies) until confluent. The cells were then induced towards neural differentiation in neural induction media (KSR and N2) comprising 15% Knockout Serum Replacement (KSR; Cat. no. 10828028; Gibco), 1% L-glutamine (100x-Gibco Cat. no. 25030081), 1% MEM (Cat. no. SH40003.01; Hyclone), and 0.1% β-mercaptoethanol (Cat. no. 31350010; Gibco) in knockout DMEM (Cat. no. 10829018; Gibco) supplemented with LDN193189 (Cat. no. 72142-1mg; Stem Cell Technologies), inhibitor BMP type 1 receptors (100 nM) and SB431542 (Cat. no. 616461-5mg; Millipore), activin receptor inhibitor (10 $\mu$M) and N2 media (8.5 mM glucose, 1× N-2 supplement [Cat. no. 17502048; Gibco] in DMEM/F12 [Cat. no. 17330-032, 1:1; Gibco]) for 12 d. After this, KSR and N2 media were replaced with N2/B27 (1:1) medium for cortical neurogenesis for up to 70 d. Both undifferentiated H9 cells and cells during differentiation were consistently treated with O-GlcNAcase inhibitor, Thiamet-G (TMG) (Cat. no. 4390-10MG; TOCRIS bioscience) dissolved in DMSO according to manufacturer's instructions at 40 mM concentration. The working concentration of TMG was kept at 40 $\mu$M throughout the study. Treatment with DMSO only was used as a control.

## Generation of hyperglycemia in rat and OGT inhibition

Wistar Rats were used for this study. Weights of the female rats were in the range of 200–240 g, and that of the male rats were 350–400 g for both control and hyperglycaemia groups. Rats were fed ad libitum with normal chow and water throughout the procedure. A 12-h dark and light cycle was maintained. For the control group, males and females were pair-mated and checked for plugs. Once the female rats were pregnant, they were kept in separate cages. For the hyperglycaemia groups, pre-gestational hyperglycaemia was induced in the female rats by intraperitoneal administration of a single bolus of streptozotocin (Cat. no. U-9889 _CAS-18883-66-4; Santa Cruz Biotechnology), 50 mg/kg of body weight (Caluwaerts et al, 2003). On the second day, glucose level was checked with the OneTouch ultra-glucometer, which is a strip-based test by adding one drop of rat blood for quantitative measurement of blood glucose levels. For most rats, the blood glucose levels reached >400 mg/dl (Table S1). Male and female rats were crossed and checked for plugs the next day. Pregnant female rats were kept in separate cages. On days E14.5, E16.5, and E18.5, different sets of control and hyperglycemic group mothers were euthanized by diethyl ether, and frontal lobe cortices of three to four embryos were dissected, pooled, snap-frozen in liquid nitrogen and stored at −80°C until further use. For OGT inhibition, pregnant hyperglycemic animals were injected with OGT inhibitor ST045849 (TimTec; 600–1,000 $\mu$g/animal) intraperitoneally from E9.5 to E13.5 as per previous report (Yu et al, 2012). The animals were euthanized at E14.5, and embryos were collected for dissection to remove brain cortices for further experiments, as explained above. All animal experiments were reviewed and approved by the Animal Ethics Committee of the UAE University (Approval numbers, ERA 2220-6057, ERA 2015 -3210).

## Protein extraction and Western blotting

During neural differentiation, the cells were saved at desired stages of differentiation and processed for protein quantification and Western blotting as follows. Cells were washed with 1× D-PBS (Cat. no. 37350; Stem Cell Technologies) gently scraped with a cell scraper (Sarstedt tissue culture scraper, Cat. no. 83.1832), collected in a tube and spun down at 1,200 rpm in a Jouan CR412 centrifuge. The supernatant was discarded, and cells were lysed in 300 $\mu$l of 1× RIPA lysis buffer (Cat. no. 9806; Cell Signalling Technologies) containing 0.1% SDS and 1X Halt protease and phosphatase inhibitor cocktail (Cat. no. 1861284; Thermo Fisher Scientific). Cells lysate was incubated on ice for 10–15 min for proper lysis followed by centrifugation at 14,000 rpm in a 5417R Eppendorf centrifuge for 10 min at 4°C. For animal tissues, embryo brain cortices dissected out at desired stages of pregnancy were snap-frozen in liquid nitrogen and stored at −80°C. On the day of protein extraction, ~100–150 $\mu$g of tissue samples were weighed out and lysed in 1× RIPA buffer containing 0.1% SDS and 1X Halt protease and phosphatase inhibitor cocktail (Cat. no. 1861284; Thermo Fisher Scientific) in a glass tube by using Pyrex glass pestle tissue homogenizer and/or ultrasonicator. Tissue lysate was collected in clean tubes and incubated on ice for 10–15 min for complete lysis followed by centrifugation at 8,000 rpm in a 5417R Eppendorf centrifuge for 30 min at 4°C. The whole lysate containing cellular proteins from cells or tissues was transferred in clean tubes. Protein quantification was carried out with Pierce BCA protein assay kit (Cat. no. 23225; Thermo Fisher Scientific) using the Infinite M200Pro multimode microplate reader (Tecan). The quantitated protein was denatured with Nupage 4× loading dye sample buffer (NP0007; Life Technologies) at 70°C for 10 min. The protein sample was aliquoted in small aliquots and stored at −80°C for further use.

The protein concentration was determined and loaded on gel according to the targeted antibody. Approximately 10–40 $\mu$g of protein was loaded on 12% or 15% of SDS–PAGE gels or Express plus 4–12% PAGE gels (Cat. no. M41210; GenScript) and electrophoresed in 1× SDS running buffer or 1× MOPS buffer (M00138; GenScript) at 100–120 V. Protein was transferred from the gel on to PVDF Transfer membrane (Cat. no. 88518; Thermo Fisher Scientific) at a constant voltage of 100 V for 1 h at 4°C. After transfer, the membrane was blocked using 5% Blotto, non-fat dry milk (Cat. no. sc-2324; Santa Cruz Biotechnology) or 5% BSA (Cat. no. A7030; Sigma-Aldrich) for 1 h at RT. The membrane was washed with 1× PBST (PBS, containing 0.05% Tween-20). According to the antibody's datasheet and references, the membrane was incubated overnight (ON) at 4°C in appropriate primary antibodies. After overnight incubation, membranes were washed with 1× PBST and probed with HRP conjugated with appropriate secondary antibodies. The information on antibodies is provided in the list of antibodies in Table S2. For visualization of protein bands, membranes were incubated with super signal West Pico and West Femto maximum sensitivity chemiluminescent substrates (Cat. nos. 34080 and 34096; Thermo Fisher Scientific) as per manufacturer instructions. The membranes were superimposed on UltraCruz autoradiography X-ray films (Cat. no. sc-201697; Santa Cruz Biotechnology), and the films were developed in GBX developer (Cat. no. 515 8613; Carestream) and fixed

in GBX fixer (Cat. no. 515 8613; Carestream) solutions. The blots were quantitated using imageJ software.

## ChIP

ChIP was performed as reported previously (Ansari et al, 2014; Ardah et al, 2018) with modifications in the sample lysis step for cells and brain tissues.

Briefly, three wells of differentiating cells from a P6 well plate were pooled and fixed with 1% formaldehyde for 10 min. For animal experiments, 50–80 mg of dissected embryo brain cortices from desired stages of pregnancy was used. The cells/ tissues were washed and lysed in cell lysis buffer. The chromatin was sheared by sonication in a Diagenode bioruptor for 10 min at 30 s on/off cycles. The lysate was immunoprecipitated using ChIP grade antibodies. The information on antibodies used is provided in Table S2. ChIP with normal IgG (Cat. no. PP64B; Millipore) was used as the negative control. ChIP'd DNA was recovered and analyzed by RT-qPCR using SYBR Green Real-Time PCR Master Mix (Thermo Fisher Scientific) in a 10-$\mu$l reaction volume. Primer sequences used to amplify promoter or coding regions were designed based on the ENCODE ChIP-seq enrichments data obtained from UCSC Genome Browser for specific histone modifications or TFs. The sequences of the primers are provided in Table S3.

## Immunoprecipitation

Co-immunoprecipitation (Co-IP) experiments were performed using previously published methods (Han et al, 2017). Briefly, the differentiating cells were scraped gently at desired stages of neural differentiation, washed twice with 1× PBS and transferred to clean tubes. Cells were lysed using freshly prepared Co-IP lysis buffer, incubated for 30 min at 4°C on a rotating platform. After incubation, lysates were centrifuged at 12,000 rpm in a 5417R Eppendorf centrifuge for 15 min at 4°C. The supernatant was transferred to a clean tube and placed on ice. Protein concentration was quantified and measured using the bicinchoninic kit (BCA protein assay kit, Cat. no. 23225; Thermo Fisher Scientific). For each IP reaction, 700 $\mu$g of protein lysate was mixed with 5 $\mu$g of EZH2 antibody. Lysate and antibody complex was incubated overnight at 4°C on a rotating platform. Post overnight incubation, 40 $\mu$l of pre-cleared Protein A agarose beads (Cat. no. 16-125; Millipore) were added in a total volume of 500 $\mu$l lysate. For negative control, a normal rabbit IgG antibody (Purified Rabbit IgG, Cat. no. PRABP01; Bio-Rad) was added in a volume of 500 $\mu$l and incubated for 2 h at 4°C. In addition, 40 $\mu$g of total protein was used as input. After incubation, the bead/antibody-protein lysate complex was washed four times with Co-IP buffer. In each wash, beads were mixed gently on a rotating platform for 3 min, the antibody/bead/lysate complex spun down using micro spin, and the supernatant was carefully discarded without disturbing the agarose beads. Beads were resuspended in 40 $\mu$l of 2× loading dye sample buffer and denatured at 95°C for 5 min and spun down after elution to separate protein complex from agarose bead. After spin, the protein complex in the supernatant was transferred to a clean tube and stored at −80°C for further use for Western blotting using anti-OGlcNAc or anti-Phospho (Ser/ Thr) antibody. The blots were quantitated using imageJ software.

## RNA-sequencing and data analysis

The cells were collected from D0, D12, D30, and D70 following induction of neural differentiation of H9 cells using the neural differentiation protocol as described above. Total RNA was isolated using MasterPure RNA Purification Kit from Epicentre (Cat. no. MCR 85102) according to the manufacturer's instructions. Libraries were prepared using NEBNext Ultra II RNA Library Prep Kit for Illumina (Cat. no. E7775; NEB) using the manufacturer's instructions. Briefly, 1 $\mu$g of total RNA was used to remove ribosomal RNAs using NEBNext rRNA Depletion Kit v2 (Human/ Mouse/Rat) (E7400L) and purified using AMPure XP beads from Beckman Coulter. Libraries were prepared on the resulting RNA samples, which included RNA fragmentation, cDNA synthesis, adapter ligation and purification as per instructions from the kit. Adapter ligated cDNA was PCR amplified using NEBNext Multiplex Oligos for Illumina (E6440S) using unique combinations of i5 and i7 index primers for multiplexing. The libraries were finally purified using AMPure XP beads and proceeded for QC analysis and sequencing. Sequencing was performed at the Genomics Core Facility, The Wistar Institute, Philadelphia, PA on Illumina Next-Generation Sequencer (NextSeq 500), followed by bioinformatic analysis.

RNA-seq data were aligned using bowtie2 (Langmead & Salzberg, 2012) algorithm against hg19 human genome version, and RSEM v1.2.12 software (Li & Dewey, 2011) was used to estimate read counts and FPKM values using gene information from Ensemble transcriptome version GRCh37.p13. Raw counts were used to estimate the significance of differential expression difference between experimental groups using DESeq2 (Love et al, 2014). Overall gene expression changes were considered significant if passed false discovery rate < 5% unless stated otherwise. Gene set enrichment analysis was carried out using QIAGEN's Ingenuity Pathway Analysis software (IPA, QIAGEN Redwood City, www.qiagen.com/ ingenuity) using "Canonical pathways," "Diseases &Functions," and "Upstream Regulators" options. Expression heat maps were generated using DESeq2 normalized count values. The significance of the overlap between stages was tested using Fisher's exact test. Hierarchical clustering was performed using Euclidean distance to visualize the expression of genes across groups that were significant in at least one inter-stage comparison with false discovery rate < 5%.

## Statistical analysis

Differentiation of hESCs into cortical neurons after treatment with TMG and solvent control was performed at least three to four times. Unless indicated otherwise, each experiment was repeated with three biological replicates and three technical replicates (real time PCR reaction) of each biological replicate for each stage of differentiation as indicated. RNA-seq experiment was performed on two biological replicates from all four stages of differentiation. Except for RNA-seq data, the results shown are mean of three biological replicates ± SD. Statistical analyses (unpaired *t* test) were made between control and treated samples individually at each time point of differentiation or developmental stage using

GraphPad Prism Software. Asterisks represent differences being significant (*$P < 0.05$, **$P < 0.01$, ***$P < 0.001$).

## Data Availability

Raw and processed RNA-seq data were deposited to the NCBI GEO database under accession number GSE169649.

### Ethics approval and consent to participate

All animal experiments were reviewed and approved by the Animal Ethics Committee of the UAE University (Approval numbers, ERA 2220-6057, ERA 2015-3210).

### Consent for publication

All authors read and corrected the manuscript and approved its final content for publication.

## Supplementary Information

## Acknowledgements

This work was supported by research grants from United Arab Emirates University (UAEU), # 31M431 (CMHS Faculty research grant), # 31R170 (Zayed Center for Health Sciences) and # 12R010 (UAEU-AUA grant) and UAE national grant, ADEK Award for Research Excellence (AARE) grant # 21M121.

### Author Contributions

S Parween: data curation, formal analysis, validation, investigation, methodology, and writing—review and editing.
TT Alawathugoda: data curation, formal analysis, investigation, and methodology.
AD Prabakaran: investigation and methodology.
ST Dheen: methodology and writing—review and editing.
RH Morse: methodology and writing—review and editing.
BS Emerald: methodology and writing—review and editing.
SA Ansari: conceptualization, resources, data curation, formal analysis, supervision, investigation, methodology, project administration, and writing—original draft, review, and editing.

### Conflict of Interest Statement

The authors declare that they have no conflict of interest.

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
