## [Reviewer comments · Life Science Alliance]

Life Science Alliance

Nutrient sensitive protein O-GlcNAcylation modulates the transcriptome during embryonic neurogenesis

Shama Parween, Thilina Alawathugoda, Ashok Daniel Prabakaran, S. Thameem Dheen, Randall H. Morse, Bright Starling Emerald, and Suraiya Ansari

DOI: <https://doi.org/10.26508/lsa.202201385>

Corresponding author(s): Suraiya Ansari, United Arab Emirates University

Review Timeline:

Submission Date:	2022-01-24
Editorial Decision:	2022-02-25
Revision Received:	2022-03-18
Editorial Decision:	2022-04-07
Revision Received:	2022-04-11
Accepted:	2022-04-11

Transaction Report:

February 25, 2022

Re: Life Science Alliance manuscript #LSA-2022-01385-T

Dr. Suraiya A Ansari
UAE University
Biochemistry and Molecular Biology
College of Medicine and Health Sciences
Tawam Hospital Campus
UAE University, Abu Dhabi 17666
UNITED ARAB EMIRATES

Dear Dr. Ansari,

Thank you for submitting your manuscript entitled "Protein O-GlcNAcylation modulates the transcriptome during embryonic neurogenesis" to Life Science Alliance. The manuscript was assessed by expert reviewers, whose comments are appended to this letter. We invite you to submit a revised manuscript addressing the Reviewer comments.

Thank you for this interesting contribution to Life Science Alliance. We are looking forward to receiving your revised manuscript.

Sincerely,

B. MANUSCRIPT ORGANIZATION AND FORMATTING:

Reviewer #1 (Comments to the Authors (Required)):

Parween et al address an important topic in their manuscript, namely how O-GlcNAcylation mechanistically affects neurogenesis. The beauty of their work is that they use human ESC-derived in vitro systems and a rat in vivo approach where they demonstrate that the alterations in neurogenesis induced by hyperglycemia can be rescued by blocking O-GlcNAcylation, hence moving from basic mechanistic insights to possible treatment options. Therefore, this work provides a significant advance with both new mechanistic insights and novel therapeutic approaches. Mechanistically, the authors demonstrate in human cells in vitro and a rat model in vivo that increased O-GlcNAcylation results in premature up-regulation of neurogenic fate determinants due to reduced H3K27me3 levels caused by modification of Ezh2 that inhibits its catalytic activity. They also provide evidence for transcriptional activation at these enhancers by PolIII Ser5 modification and increased K112-O-GlcNAcylation. These data are convincing, but require further information about sample size and statistical tests. With these additions the manuscript is ready for publication at Life Science Alliance.

Specific suggestions:

- 1) The authors should depict the individual biological replicate data points in each of the histograms throughout the manuscript.
- 2) The number of biological replicates must be minimum 3 to apply statistics. For Figure 2 the authors write that they used 2-3 biological replicates, but 2 is too low.
- 3) I do not understand what the authors are referring to when writing - as for Figure 2: The data are representative of 3 independent biological replicates. For the histograms shown in Figure 2B they should show the mean or median from the biological replicates, not a representative example meaning from 1 experiment. I guess the latter is also the reason why there are no statistical tests used in Figure 2, but we need statistical analysis to evaluate what is a significant difference and what not - at least for the core results. The same comments apply to Figure 5 and 6 and 7 and 8.
- 4) In Figure 3 the authors use two-tailed unpaired students t-test, but this is only possible if the data are normally distributed. Did the authors test normal distribution of the data. If so, please indicate which test was used. If not, use a non-parametric test.
- 5) In the text of the manuscript please specify and discriminate significant differences from "trends" that were not tested by statistics. For example, on p.17 the authors write that "the levels of H3K27me3 remained lower at these stages (E16.5 and 18.5) in hyperglycemia except for Foxg1 and Ngn2, which showed up-regulation at E16.5...." However, none of these statements is corroborated by a statistical test and indeed looking at Figure 5A I cannot see why Foxg1 is claimed to be up-regulated with fully overlapping STD between control and treatment, while Eomes is claimed not to be up-regulated at E16.5 which has as overlapping standard deviation.
- 6) What is the authors' interpretation of Oct4 up-regulation in the TMG-treated samples at day 30 and 70 of differentiation (last line on p.11)? Does this manifest at protein level? Would it disturb neuronal differentiation?

Reviewer #2 (Comments to the Authors (Required)):

The manuscript by Parween et al explores the effect of O-GlcNAcylation (O-GlcNAc) on epigenetic regulation of neural development taking advantage of an in vitro and in vivo experimental system. For the analysis in vitro the authors investigate human neurons derived from embryonic stem cells differentiating in the presence or absence of TMG to upregulate O-GlcNAc. Using RNA seq they investigated treatment-dependent changes in gene transcription at different stages on neural differentiation. The authors conclude that culture differentiating in the presence of TGM show increased in gene expression, which they propose is a consequence of transcriptional de-repression as indicated by changes in histone modifications and activity of RNA polymerase II. For their in vivo analysis they show that streptozotocin induced hyperglycemia in pregnant rats caused increased O-GlcNAc and similar epigenetic dysregulation in the brain, which was restored by pharmacological inhibition of O-GlcNAc.

The overall goal of the study is interesting and poorly investigated in the literature. However, as detailed below, some concerns about the experimental procedure, the interpretation of the data and the essentially correlative nature of the evidence presented limit my enthusiasm for the study.

- 1) Using the same experimental system in vitro the authors have shown that the treatment not only promotes in time neuronal

differentiation, but also leads to significant changes in cell death and proliferation. In light of this an increase in the expression of neuronal genes should not be surprising especially given that most of the changes are observed at later differentiation stages. Prove of increased gene expression should be provided by analysis of protein expression at cellular level.

2) The fact that transcriptional modulation essentially involves an increase in transcription levels, per se does not represent a prove of transcriptional de-repression. Therefore, the data showing changes in transcriptional repression are critical to support this hypothesis. Instead, the data shown in figure 2B illustrate a lot of variability in EZH2 promoter occupancy, which hardly appears significant. Whereas the data concerning H3K27me3 and H3K4me3 appear more promising, the statistical significance of the data is not provided.

3) The description of the results in this paragraph is also ambiguous. The changes in EZH2 and H3K4me3 are first described as if they were significant, but at end of the chapter it is mentioned that only H3K27me3 is significantly affected. A similar ambiguity concerns the description of experiments in Fig. 3A, B where no significant change in EZH2 O-GlcNAc is observed.

4) The change in Histone 3 could be associated also to the toxic effect of TGM. This possibility should be at least discussed.

5) The suggestion that changes in epigenetic modification could be due to the increase in EZH2-Thr311 is based only on correlative evidence.

6) The experiments in the in vivo model show some parallel with the in vitro model in terms of changes in differentiation and proliferation and epigenetic. However, no immunohistochemistry is shown to illustrate changes in cortical development and how this is affected by the various treatments. Moreover, statistical analysis is missing from figure 5.

7) Also in the in vivo experimental paradigm, the link between epigenetic changes and alteration in differentiation is purely correlative whereas the extent to which increased O-GlcNAc mediates the effect of hyperglycemia remains poorly investigated. Although, the OGT inhibitor rescued the effect of hyperglycemia on EZH2-Thr311 no changes are observed in H3K27me3 and H3K4me3. What about the effect of the OGT inhibitor on neurodevelopment (neuronal and stem cell marker expression)?

Response to reviewers' comments

We thank the editors and reviewers for carefully reading our manuscript and for their constructive comments and suggestions which we have incorporated in the revised manuscript. We believe that the incorporation of these changes has made our manuscript suitable for publication in the Life Science Alliance.

Point wise response to reviewers' comments are given below:

Reviewer's comments

Reviewer #1 (Comments to the Authors (Required)):

Parween et al address an important topic in their manuscript, namely how O-GlcNAcylation mechanistically affects neurogenesis. The beauty of their work is that they use human ESC-derived in vitro systems and a rat in vivo approach where they demonstrate that the alterations in neurogenesis induced by hyperglycemia can be rescued by blocking O-GlcNAcylation, hence moving from basic mechanistic insights to possible treatment options. Therefore, this work provides a significant advance with both new mechanistic insights and novel therapeutic approaches. Mechanistically, the authors demonstrate in human cells in vitro and a rat model in vivo that increased O-GlcNAcylation results in premature up-regulation of neurogenic fate determinants due to reduced H3K27me3 levels caused by modification of Ezh2 that inhibits its catalytic activity. They also provide evidence for transcriptional activation at these enhancers by PolII Ser5 modification and increased K112-O-GlcNAcylation. These data are convincing, but require further information about sample size and statistical tests. With these additions the manuscript is ready for publication at Life Science Alliance.

Response: We thank the reviewer for critically reading of our manuscript and insightful suggestions. We also apologize that our wording in several places has confused the reviewer about sample size and statistical analysis of the data. We have carefully addressed the points raised by the reviewer (please see below) and have incorporated necessary changes in the text and figures.

Specific suggestions:

1) The authors should depict the individual biological replicate data points in each of the histograms throughout the manuscript.

Response: Thanks for the suggestion! We have re-plotted each of the histograms which now include individual data points in the revised manuscript (please see revised figures and text).

2) The number of biological replicates must be minimum 3 to apply statistics. For Figure 2 the authors write that they used 2-3 biological replicates, but 2 is too low.

Response: The data represent mean of three biological replicates (samples collected from three independent hESCs-neuronal differentiation experiment) for all the data throughout the manuscript except the RNA-seq experiments which were from two biological replicates.

3) I do not understand what the authors are referring to when writing - as for Figure 2: The data are representative of 3 independent biological replicates. For the histograms shown in Figure 2B they should

show the mean or median from the biological replicates, not a representative example meaning from 1 experiment. I guess the latter is also the reason why there are no statistical tests used in Figure 2, but we need statistical analysis to evaluate what is a significant difference and what not - at least for the core results. The same comments apply to Figure 5 and 6 and 7 and 8.

Response: The data shown in figures, 2B, 5, 6, 7 and 8 are all mean of three biological replicates (with 3 technical repeats) \pm SD. Individual data points for all these plots have been included as suggested by the reviewer.

4) In Figure 3 the authors use two-tailed unpaired students t-test, but this is only possible if the data are normally distributed. Did the authors test normal distribution of the data. If so, please indicate which test was used. If not, use a non-parametric test.

Response: Our apologies to the reviewer as we are unsure of what other tests would be better. Please note that the differences are measured between two groups only (control vs TMG treated from each stage of differentiation in cell culture (Fig 3) and control vs hyperglycemia or HG vs ST in in-vivo (Fig 9) for these experiments, and so a two-tailed Student's t-test was used. We had used same statistics for western quantitation in our previous publications [1, 2] and same tests were applied by other authors for this type of experiments for example, [3]. For all statistical analysis, we had used GraphPad Prism Software.

5) In the text of the manuscript please specify and discriminate significant differences from "trends" that were not tested by statistics. For example, on p.17 the authors write that "the levels of H3K27me3 remained lower at these stages (E16.5 and 18.5) in hyperglycemia except for Foxg1 and Ngn2, which showed up-regulation at E16.5...." However, none of these statements is corroborated by a statistical test and indeed looking at Figure 5A I cannot see why Foxg1 is claimed to be up-regulated with fully overlapping STD between control and treatment, while Eomes is claimed not to be up-regulated at E16.5 which has as overlapping standard deviation.

Response: Thank you for the comments! We agree with the reviewer and therefore, for these experiments, we have included p-values. Individual data points are also included for all of these plots. We have also re-phrased the description on these data (page 18, lines 404-413). For Fig 5A, it is rightly pointed out by the reviewer that the changes in both Eomes and Foxg1 are similar but both are not statistically significant (please see the updated Fig 5 in the revised manuscript).

6) What is the authors' interpretation of Oct4 up-regulation in the TMG-treated samples at day 30 and 70 of differentiation (last line on p.11)? Does this manifest at protein level? Would it disturb neuronal differentiation?

Response: We had performed western blotting for OCT4 and NANOG on D30 and D70 of neural differentiation and found that although OCT4 and NANOG express on D30 and to a lesser extent on D70 in TMG, the levels are much lower than what is normally expressed in pluripotent cells (data not shown). We believe that these genes may get de-repressed due to elevated O-GlcNAc levels but possibly in a fraction of cells. We plan to investigate this further in our future study.

Reviewer #2 (Comments to the Authors (Required)):

The manuscript by Parween et al explores the effect of O-GlcNAcylation (O-GlcNAc) on epigenetic regulation of neural development taking advantage of an in vitro and in vivo experimental system. For the analysis in-vitro the authors investigate human neurons derived from embryonic stem cells differentiating in the presence or absence of TMG to upregulate of O-GlcNAc. Using RNA seq they investigated treatment-dependent changes in gene transcription at different stages on neural differentiation. The authors conclude that culture differentiating in the presence of TGM show increased in gene expression, which they propose is a consequence of transcriptional de-repression as indicated by changes in histone modifications and activity of RNA polymerase II. For their in vivo analysis they show that streptozotocin induced hyperglycemia in pregnant rats caused increased O-GlcNAc and similar epigenetic dysregulation in the brain, which was restored by pharmacological inhibition of O-GlcNAc.

The overall goal of the study is interesting and poorly investigated in the literature. However, as detailed below, some concerns about the experimental procedure, the interpretation of the data and the essentially correlative nature of the evidence presented limit my enthusiasm for the study.

1) Using the same experimental system in vitro the authors have shown that the treatment not only promotes in time neuronal differentiation, but also leads to significant changes in cell death and proliferation. In light of this an increase in the expression of neuronal genes should not be surprising especially given that most of the changes are observed at later differentiation stages. Prove of increased gene expression should be provided by analysis of protein expression at cellular level.

Response: We have investigated the effect of increased O-GlcNAcylation on some of the targets at protein levels in our previous work [2] which included SOX2, PAX6, OTX2 (show reduced expression due to TMG) and TBR1 (which shows increased expression). However, in the present study, our major goal was to understand epigenetic/transcriptional regulatory mechanisms mediated by protein O-GlcNAcylation during neurogenesis. Therefore, RNA expression and ChIP were mainly used for this. In the case of in-vivo experiments though, we did check the protein levels of many of the same targets in hyperglycemia (Figure 4 and supplementary Figure S2).

2) The fact that transcriptional modulation essentially involves an increase in transcription levels, per se does not represent a prove of transcriptional de-repression. Therefore, the data showing changes in transcriptional repression are critical to support this hypothesis. Instead, the data shown in figure 2B illustrate a lot of variability in EZH2 promoter occupancy, which hardly appears significant. Whereas the data concerning H3K27me3 and H3K4me3 appear more promising, the statistical significance of the data is not provided.

Response: We have modified the text to emphasize that little change in EZH2 association is observed upon TMG treatment, while levels of H3K27me3 are decreased and levels of H3K4me3 are increased, consistent with the observed changes in transcript levels. We now include statistical significance for these experiments. The apparent discrepancy between EZH2 levels and H3K27me3 is explored in the next section, in which we report increased EZH2-Thr311p caused by TMG and hyperglycemia. It is important to note that, similar to our observation, a previous study has reported that EZH2-Thr311p is recruited to H3 at comparable levels to non-phosphorylated form [4].

3) The description of the results in this paragraph is also ambiguous. The changes in EZH2 and H3K4me3 are first described as if they were significant, but at end of the chapter it is mentioned that only H3K27me3 is significantly affected. A similar ambiguity concerns the description of experiments in Fig. 3A, B where no significant change in EZH2 O-GlcNAc is observed.

Response: Our apologies for this ambiguity. The levels of both H2K27me3 and H3K4me3 are significantly affected both in cell culture and in-vivo experiments (individual data points and p-values are included on the plots in the revised figures) whereas EZH2 recruitment was not affected significantly on most of these promoters. This issue could be due to our wording being confusing in several places. We have taken a note of this and re-phrased several sentences (please see the revised manuscript) to clarify these points. Similarly, for the description of experiments in Fig. 3A, B, we have added a sentence (page 15, lines 333-334) to clarify these results.

4) The change in Histone 3 could be associated also to the toxic effect of TGM. This possibility should be at least discussed.

Response: We believe the changes in histone H3 could not have been due to toxicity of TMG rather it could be the result of increased O-GlcNAcylation because H3 levels were similarly downregulated on these promoters due to maternal hyperglycemia as well in in-vivo experiments. Similarly, the levels of H2B were also reduced on several of these promoters especially on D30 of in-vitro cell culture (Fig 6A) and in hyperglycemic embryos on E14.5 and E18.5 (Fig 7E). These results may suggest that hyperglycemia and elevated O-GlcNAc result in increased nucleosomal depletion at chromatin however further experiments such as ATAC-seq are required to verify these claims.

5) The suggestion that changes in epigenetic modification could be due to the increase in EZH2-Thr311 is based only on correlative evidence.

Response: The reviewer is right that the suggestions that changes in H3K37me3/H3K4me3 levels could be due to the increase in EZH2-Thr311p is based only on correlative evidence. However, this correlation is based on a strong evidence from a previous study [4] where authors have thoroughly investigated the role of EZH2-Thr311p by using phosphorylation-deficient T311A mutant and the phosphorylation mimetic mutant, T311E-EZH2 in various cancer cell lines. This led to the identification of the inhibitory role of EZH2-Thr311p on H3K27 methylation. Similar experiments are needed to be performed in normal cells during hESCs-neuronal differentiation as well as on animal cells grown in high glucose. As the reviewer will agree that this will be a large independent investigation and is outside the scope of this study.

6) The experiments in the in vivo model show some parallel with the in vitro model in terms of changes in differentiation and proliferation and epigenetic. However, no immunohistochemistry is shown to illustrate changes in cortical development and how this is affected by the various treatments. Moreover, statistical analysis is missing from figure 5.

Response: In-vivo experiments in this study were mainly performed to validate O-GlcNAc mediated gene regulatory mechanisms identified in cell culture experiments. Phenotypic characterizations of hyperglycemic animals were therefore not planned. However; a previous study has shown a thorough immunohistochemistry on the developing brains of hyperglycemic mice and found similar defects in embryonic neurogenesis [5] which we have found through western blotting of neuronal markers in our rat model. For our future investigations, we are planning phenotypic characterizations of developing brains in hyperglycemia and the effects of OGT inhibition both embryonic and postnatal as well as further molecular characterizations at genome-wide RNA and epigenetic levels in animals.

7) Also in the in vivo experimental paradigm, the link between epigenetic changes and alteration in differentiation is purely correlative whereas the extent to which increased O-GlcNAc mediates the effect of hyperglycemia remains poorly investigated. Although, the OGT inhibitor rescued the effect of hyperglycemia on EZH2-Thr311 no changes are observed in H3K27me3 and H3K4me3. What about the effect of the OGT inhibitor on neurodevelopment (neuronal and stem cell marker expression)?

Response: We agree with the reviewer that OGT inhibitor did not cause a global change in H3K27me3 and H3K4me3 levels observed through western blotting however a reduced H3K27me3 and increased H3K4me3 in hyperglycemia through ChIP is indicative of epigenetic modulations. As inquired by the reviewer, we performed western blotting for neuronal markers, Eomes, Tbr1 and β -Tubulin III in HG and ST compared to control and found that OGT inhibition does normalize the upregulation of these markers due to hyperglycemia. This data has been included as supplementary Fig S2 in the revised Manuscript.

1. Ardah, M.T., et al., *Saturated fatty acid alters embryonic cortical neurogenesis through modulation of gene expression in neural stem cells*. J Nutr Biochem, 2018. **62**: p. 230-246.
2. S, P., et al., *Higher O-GlcNAc Levels Are Associated with Defects in Progenitor Proliferation and Premature Neuronal Differentiation during in-Vitro Human Embryonic Cortical Neurogenesis*. Frontiers in cellular neuroscience, 2017. **11**.
3. Cai, S.Y., et al., *Hepatic NFAT signaling regulates the expression of inflammatory cytokines in cholestasis*. J Hepatol, 2021. **74**(3): p. 550-559.
4. Wan, L., et al., *Phosphorylation of EZH2 by AMPK Suppresses PRC2 Methyltransferase Activity and Oncogenic Function*. Mol Cell, 2018. **69**(2): p. 279-291.e5.
5. Ji, S., et al., *Maternal hyperglycemia disturbs neocortical neurogenesis via epigenetic regulation in C57BL/6J mice*. Cell Death Dis, 2019. **10**(3): p. 211.

April 7, 2022

RE: Life Science Alliance Manuscript #LSA-2022-01385-TR

Dr. Suraiya A Ansari
United Arab Emirates University
Biochemistry and Molecular Biology
College of Medicine and Health Sciences
Tawam Hospital Campus
UAE University, Abu Dhabi 17666
United Arab Emirates

Dear Dr. Ansari,

Thank you for submitting your revised manuscript entitled "Nutrient sensitive protein O-GlcNAcylation modulates the transcriptome during embryonic neurogenesis". We would be happy to publish your paper in Life Science Alliance pending final revisions necessary to meet our formatting guidelines.

- please upload your supplementary figures as single files and add supplementary figure legends to the main manuscript text (after the legends for the main figures); the tables can remain in one file but must be in doc or excel format
- please add Keywords and Category to our manuscript system
- please add the Twitter handle of your host institute/organization as well as your own or/and one of the authors in our system
- please make sure that the author names in our system match the author names in your manuscript
- please use the [10 author names, et al.] format in your references (i.e. limit the author names to the first 10)
- please add a callout to Figure 10 in your main manuscript text
- please rename the Experimental Procedures section to Materials & Methods
- please add molecular weights next to all blots

A. FINAL FILES:

B. MANUSCRIPT ORGANIZATION AND FORMATTING:

Sincerely,

Reviewer #1 (Comments to the Authors (Required)):

The authors have addressed all my comments, but should definitely contact a statistician to either use a non-parametric test or a posthoc test.

Reviewer #2 (Comments to the Authors (Required)):

The authors have replied to the issues I have raised during the first round of revision.

April 11, 2022

RE: Life Science Alliance Manuscript #LSA-2022-01385-TRR

Dr. Suraiya A Ansari
United Arab Emirates University
Biochemistry and Molecular Biology
College of Medicine and Health Sciences
Tawam Hospital Campus
UAE University, Abu Dhabi 17666
United Arab Emirates

Dear Dr. Ansari,

Thank you for submitting your Research Article entitled "Nutrient sensitive protein O-GlcNAcylation modulates the transcriptome during embryonic neurogenesis". It is a pleasure to let you know that your manuscript is now accepted for publication in Life Science Alliance. Congratulations on this interesting work.

DISTRIBUTION OF MATERIALS:

Again, congratulations on a very nice paper. I hope you found the review process to be constructive and are pleased with how the manuscript was handled editorially. We look forward to future exciting submissions from your lab.

Sincerely,
